# Flattening a Hierarchical Clustering through Active Learning

**Fabio Vitale**
Department of Computer Science
INRIA Lille, France &
Sapienza University of Rome, Italy
`fabio.vitale@inria.fr`

**Anand Rajagopalan**
Google Research NY
New York, USA
`anandbr@google.com`

**Claudio Gentile**
Google Research NY
New York, USA
`cgentile@google.com`

## Abstract

We investigate active learning by pairwise similarity over the leaves of trees originating from hierarchical clustering procedures. In the realizable setting, we provide a full characterization of the number of queries needed to achieve perfect reconstruction of the tree cut. In the non-realizable setting, we rely on known important-sampling procedures to obtain regret and query complexity bounds. Our algorithms come with theoretical guarantees on the statistical error and, more importantly, lend themselves to *linear-time* implementations in the relevant parameters of the problem. We discuss such implementations, prove running time guarantees for them, and present preliminary experiments on real-world datasets showing the compelling practical performance of our algorithms as compared to both passive learning and simple active learning baselines.

## 1 Introduction

Active learning is a learning scenario where labeled data are scarce and/or expensive to gather, as they require careful assessment by human labelers. This is often the case in several practical settings where machine learning is routinely deployed, from image annotation to document classification, from speech recognition to spam detection, and beyond. In all such cases, an active learning algorithm tries to limit human intervention by seeking as little supervision as possible, still obtaining accurate prediction on unseen samples. This is an attractive learning framework offering substantial practical benefits, but also presenting statistical and algorithmic challenges.

A main argument that makes active learning effective is when combined with methods that exploit the *cluster structure* of data (e.g., [11, 21, 10], and references therein), where a cluster typically encodes some notion of semantic similarity across the involved data points. An ubiquitous solution to clustering is to organize data into a *hierarchy*, delivering clustering solutions at different levels of resolution. An (agglomerative) Hierarchical Clustering (HC) procedure is an unsupervised learning method parametrized by a similarity function over the items to be clustered and a linkage function that lifts similarity from items to clusters of items. Finding the "right" level of resolution amounts to turning a given HC into a *flat* clustering by cutting the resulting tree appropriately. We would like to do so by resorting to human feedback in the form of *pairwise similarity* queries, that is, yes/no questions of the form "are these two products similar to one another ?" or "are these two news items covering similar events ?". It is well known that such queries are relatively easy to respond to, but are also intrinsically prone to subjectiveness and/or noise. More importantly, the hierarchy at hand need not be aligned with the similarity feedback we actually receive.

In this paper, we investigate the problem of cutting a tree originating from a pre-specified HC procedure through pairwise similarity queries generated by active learning algorithms. Since the tree is typically not consistent with the similarity feedback, that is to say, the feedback is *noisy*, we are lead to tackle this problem under a variety of assumptions about the nature of this noise (from noiseless to random but persistent to general agnostic). Moreover, because different linkage functions

applied to the very same set of items may give rise to widely different tree topologies, our study also focuses on characterizing active learning performance as a function of the structure of the tree at hand. Finally, because these hierarchies may in practice be sizeable (in the order of billion nodes), scalability will be a major concern in our investigation.

**On motivation.** It is often the case in big organizations that data processing pipelines are split into *services*, making Machine Learning solution providers be constrained by the existing hardware/software infrastructure. In the Active Learning applications that motivate this work, the hierarchy over the items to be clustered is provided by a third party, i.e., by an exogenous data processing tool that relies on side information on the items (e.g., word2vec mappings and associated distance functions) which are possibly generated by yet another service, etc. In this modular environment, it is reasonable to assume that the tree is *given to us as part of the input* of our Active Learning problem. The human feedback our algorithms rely upon may or may not be consistent with the tree at hand both because human feedback is generally noisy and because this feeback may originate from yet another source of data, e.g., another production team in the organization that was not in charge of building the original tree. In fact, the same tree over the data items may serve the clustering needs of different groups within the organization, having different goals and views on the same data. This also motivates why, when studying this problem, we are led to consider different noise scenarios, that is, the presence of noisy feedback and the possibility that the given tree is not consistent with the clustering corresponding to the received feedbacks.

**Our contribution.** In the realizable setting (both noiseless and persistent noisy, Section 3), we introduce algorithms whose expected number of queries scale with the *average complexity* of tree cuts, a notion which is introduced in this paper. A distinctive feature of these algorithms is that they are rather ad hoc in the way they deal with the structure of our problem. In particular, they cannot be seen as finding the query that splits the version space as evenly as possible, a common approach in many active learning papers (e.g., [12, 25, 15, 16, 27, 24], and references therein). We then show that, at least in the noiseless case, this average complexity measure characterizes the expected query complexity of the problem. Our ad hoc analyses are beneficial in that they deliver sharper guarantees than those readily available from the above papers. In addition, and perhaps more importantly for practical usage, our algorithms admit *linear-time* implementations in the relevant parameters of the problem (like the number of items to be clustered). In the non-realizable setting (Section 4), we build on known results in importance-weighted active learning (e.g., [5, 6]) to devise a selective sampling algorithm working under more general conditions. While our statistical analysis follows by adapting available results, our goal here is to rather come up with fast implementations, so as to put the resulting algorithms on the same computational footing as those operating under (noisy) realizability assumptions. By leveraging the specific structure of our hypothesis space, we design a fast incremental algorithm for selective sampling whose running time per round is linear in the height of the tree. In turn, this effort paves the way for our experimental investigation (Section 5), where we compare the effectiveness of the two above-mentioned approaches (realizable with persistent noise vs non-realizable) on real data originating from various linkage functions. Though preliminary in nature, these experiments seem to suggest that in practice the algorithms originating from the persistent noise assumption exhibit more attractive learning curves than those working in the more general non-realizable setting.

**Related work.** The literature on active learning is vast, and we can hardly do it justice here. In what follows we confine ourselves to the references which we believe are closest to our paper. Since our sample space is discrete (the set of all possible pairs of items from a finite set of size $n$), our realizable setting is essentially a *pool-based* active learning setting. Several papers have considered greedy algorithms which generalize binary search [1, 20, 12, 25, 15, 24]. The query complexity can be measured either in the worst case or averaged over a prior distribution over all possible labeling functions in a given set. The query complexity of these algorithms can be analyzed by comparing it to the best possible query complexity achieved for that set of items. In [12] it is shown that if the probability mass of the version space is split as evenly as possible then the approximation factor for its average query complexity is $\mathcal{O}(\log(1/p_m))$, where $p_m$ is the minimal prior probability of any considered labeling function. [15] extended this result through a more general approach to approximate greedy rules, but with the worse factor $\mathcal{O}(\log^2(1/p_m))$. [20] observed that modifying the prior distribution always allows one to replace $\mathcal{O}(\log(1/p_m))$ by the smaller factor $\mathcal{O}(\log N)$, where $N$ is the size of the set of labeling functions. Results of a similar flavor are contained in [25, 24]. In our case, $N$ can be exponential in $n$ (see Section 2), making these landmark results too broad to be tight for our specific setting. Furthermore, some of these papers (e.g., [12, 25, 24]) have

only theoretical interest because of their difficult algorithmic implementation. Interesting advances on this front are contained in the more recent paper [27], though when adapted to our specific setting, their results give rise to worse query bounds than ours. In the same vein are the papers by [7, 8], dealing with persistent noise. Finally, in the non-realizable setting, our work fully relies on [6], which in turns builds on standard references like [9, 5, 17] – see, e.g., the comprehensive survey by [18]. Further references, specifically related to clustering with queries, are mentioned in Appendix A.

## 2 Preliminaries and learning models

We consider the problem of finding cuts of a given binary tree through pairwise similarity queries over its leaves. We are given in input a binary[1] tree $T$ originating from, say, an agglomerative (i.e., bottom-up) HC procedure (single linkage, complete linkage, etc.) applied to a set of items $L = \{x_1, \ldots, x_n\}$. Since $T$ is the result of successive (binary) merging operations from bottom to top, $T$ turns out to be a *strongly binary tree*[2] and the items in $L$ are the leaves of $T$. We will denote by $V$ the set of nodes in $T$, including its leaves $L$, and by $r$ the root of $T$. The height of $T$ will be denoted by $h$. When referring to a subtree $T'$ of $T$, we will use the notation $V(T'), L(T'), r(T')$, and $h(T')$, respectively. We also denote by $T(i)$ the subtree of $T$ rooted at node $i$, and by $L(i)$ the set of leaves of $T(i)$, so that $L(i) = L(T(i))$, and $r(T(i)) = i$. Moreover, $\mathrm{par}(i)$ will denote the parent of node $i$ (in tree $T$), $\mathrm{left}(i)$ will be the left-child of $i$, and $\mathrm{right}(i)$ its right child.

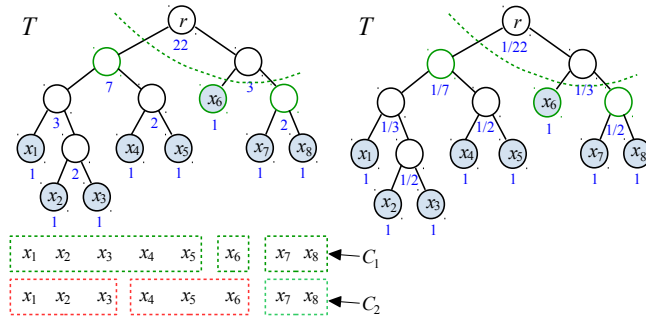

**Figure 1: Left:** A binary tree corresponding to a hierarchical clustering of the set of items $L = \{x_1, \ldots, x_8\}$. The cut depicted in dashed green has two nodes above and the rest below. This cut induces over $L$ the flat clustering $C_1 = \{\{x_1, x_2, x_3, x_4, x_5\}, \{x_6\}, \{x_7, x_8\}\}$ corresponding to the leaves of the subtrees rooted at the 3 green-bordered nodes just below the cut (the lower boundary of the cut). Clustering $C_1$ is therefore realized by $T$. On the contrary, clustering $C_2 = \{\{x_1, x_2, x_3\}, \{x_4, x_5, x_6\}, \{x_7, x_8\}\}$ is not. Close to each node $i$ is also displayed the number $N(i)$ of realized cuts by the subtree rooted at $i$. For instance, in this figure, $7 = 1 + 3 \cdot 2$, and $22 = 1 + 7 \cdot 3$, so that $T$ admits overall $N(T) = 22$ cuts. **Right:** The same figure, where below each node $i$ are the probabilities $\boldsymbol{p}(i)$ encoding a uniform prior distribution over cuts. Notice that $\boldsymbol{p}(i) = 1/N(i)$ so that, like all other cuts, the depicted green cut has probability $(1 - 1/22) \cdot (1 - 1/3) \cdot (1/7) \cdot 1 \cdot (1/2) = 1/22$.

A *flat* clustering $\mathcal{C}$ of $L$ is a partition of $L$ into disjoint (and non-empty) subsets. A *cut* $c$ of $T$ of size $K$ is a set of $K$ edges of $T$ that partitions $V$ into two disjoint subsets; we call them the nodes *above* $c$ and the nodes *below* $c$. Cut $c$ also univocally induces a clustering over $L$, made up of the clusters $L(i_1), L(i_2), \ldots, L(i_K)$, where $i_1, i_2, \ldots, i_K$ are the nodes below $c$ that the edges of $c$ are incident to. We denote this clustering by $\mathcal{C}(c)$, and call the nodes $i_1, i_2, \ldots, i_K$ the *lower boundary* of $c$. We say that clustering $\mathcal{C}_0$ is *realized* by $T$ if there exists a cut $c$ of $T$ such that $\mathcal{C}(c) = \mathcal{C}_0$. See Figure 1 (left) for a pictorial illustration.

Clearly enough, for a given $L$, and a given tree $T$ with set of leaves $L$, not all possible clusterings over $L$ are realized by $T$, as the number and shape of the clusterings realized by $T$ are strongly influenced by $T$'s structure. Let $N(T)$ be the number of clusterings realized by $T$ (notice that this is also equal to the number of distinct cuts admitted by $T$). Then $N(T)$ can be computed through a simple recursive formula. If we let $N(i)$ be the number of cuts realized by $T(i)$, one can easily verify that $N(i) = 1 + N(\mathrm{left}(i)) \cdot N(\mathrm{right}(i))$, with $N(x_i) = 1$ for all $x_i \in L$. With this notation, we then have $N(T) = N(r(T))$. If $T$ has $n$ leaves, $N(T)$ ranges from $n$, when $T$ is a degenerate line tree, to the exponential $\lfloor \alpha^n \rfloor$, when $T$ is the full binary tree, where $\alpha \simeq 1.502$ (e.g., http://oeis.org/A003095). See again Figure 1 (left) for a simple example.

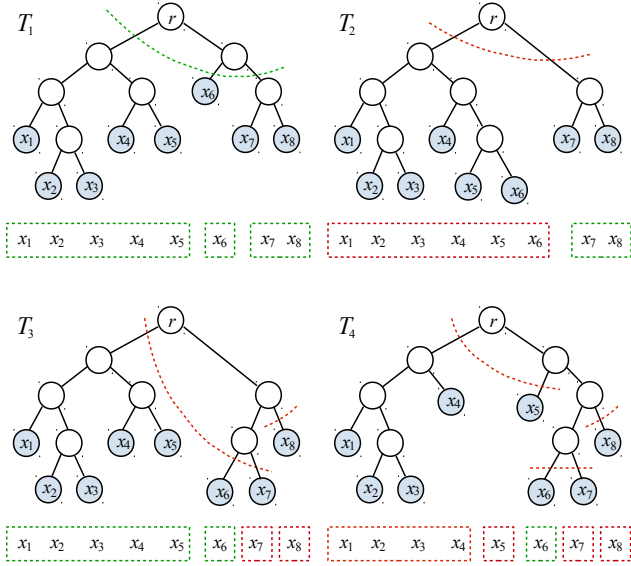

**Figure 2:** Cuts and corresponding values of $d_H(\Sigma, \widehat{\mathcal{C}})$ in four input trees for the same ground-truth $\Sigma$ determined by the clustering $\{\{x_1, x_2, x_3, x_4, x_5\}, \{x_6\}, \{x_7, x_8\}\}$. Underneath each tree is the clustering $\widehat{\mathcal{C}}$ induced by the depicted cut. The color of a cluster is green if it corresponds to a cluster in the clustering of $\Sigma$, and is red otherwise. In $T_1$ we have $d_H(\Sigma, \widehat{\mathcal{C}}) = 0$ (realizable case). The other 3 trees illustrate the non-realizable case with the cut minimizing $d_H(\Sigma, \widehat{\mathcal{C}})$ on the corresponding tree. Recalling that $d_H(\Sigma, \widehat{\mathcal{C}})$ counts *ordered* pairs, in $T_2$ we have $d_H(\Sigma, \widehat{\mathcal{C}}) = 10$, because $x_6$ now belongs to the first cluster according to clustering $\widehat{\mathcal{C}}$. In $T_3$ we have $d_H(\Sigma, \widehat{\mathcal{C}}) = 2$. Finally, in $T_4$ it is easy to verify that $d_H(\Sigma, \widehat{\mathcal{C}}) = 10$.

A *ground-truth* matrix $\Sigma$ is an $n \times n$ and $\pm1$-valued symmetric matrix $\Sigma = [\sigma(x_i, x_j)]_{i,j=1}^{n \times n}$ encoding a pairwise similarity relation over $L$. Specifically, if $\sigma(x_i, x_j) = 1$ we say that $x_i$ and $x_j$ are similar, while if $\sigma(x_i, x_j) = -1$ we say they are dissimilar. Moreover, we always have $\sigma(x_i, x_i) = 1$ for all $x_i \in L$. Notice that $\Sigma$ need not be consistent with a given clustering over $L$, i.e., the binary relation defined by $\Sigma$ over $L$ need not be transitive.

Given $T$ and its leaves $L$, an active learning algorithm $A$ proceeds in a sequence of rounds. In a purely active setting, at round $t$, the algorithm queries a pair of items $(x_{i_t}, x_{j_t})$, and observes the associated label $\sigma(x_{i_t}, x_{j_t})$. In a *selective sampling* setting, at round $t$, the algorithm is presented with $(x_{i_t}, x_{j_t})$ drawn from some distribution over $L \times L$, and has to decide whether or not to query the associated label $\sigma(x_{i_t}, x_{j_t})$. In both cases, the algorithm is stopped at some point, and is compelled to commit to a specific cut of $T$ (inducing a flat clustering over $L$). Coarsely speaking, the goal of $A$ is to come up with a good cut of $T$, by making as few queries as possible on the entries of $\Sigma$.

**Noise Models.** The simplest possible setting, called *noiseless realizable* setting, is when $\Sigma$ itself is consistent with a given clustering realized by $T$, i.e., when there exists a cut $c^*$ of $T$ such that $\mathcal{C}(c^*) = \{L(i_1), \ldots, L(i_K)\}$, for some nodes $i_1, \ldots, i_K \in V$, that satisfies the following: For all $r = 1, \ldots, K$, and for all pairs $(x_i, x_j) \in L(i_r) \times L(i_r)$ we have $\sigma(x_i, x_j) = 1$, while for all other pairs we have $\sigma(x_i, x_j) = -1$. We call (persistent) *noisy realizable* setting one where $\Sigma$ is generated as follows. Start off from the noiseless ground-truth matrix, and call it $\Sigma^*$. Then, in order to obtain $\Sigma$ from $\Sigma^*$, consider the set of all $\binom{n}{2}$ pairs $(x_i, x_j)$ with $i < j$, and pick uniformly at random a subset of size $\lfloor \lambda \binom{n}{2} \rfloor$, for some $\lambda \in [0, 1/2)$. Each such pair has flipped label in $\Sigma$: $\sigma(x_i, x_j) = 1 - \sigma^*(x_i, x_j)$. This is then combined with the symmetric $\sigma(x_i, x_j) = \sigma(x_j, x_i)$, and the reflexive $\sigma(x_i, x_i) = 1$ conditions. We call $\lambda$ the noise level. Notice that this kind of noise is random but *persistent*, in that if we query the same pair $(x_i, x_j)$ twice we do obtain the same answer $\sigma(x_i, x_j)$. Clearly, the special case $\lambda = 0$ corresponds to the noiseless setting. Finally, in the general *non-realizable* (or agnostic) setting, $\Sigma$ is an arbitrary matrix that need not be consistent with any clustering over $L$, in particular, with any clustering over $L$ realized by $T$.

**Error Measure.** If $\Sigma$ is some ground-truth matrix over $L$, and $\widehat{c}$ is the cut output by $A$, with induced clustering $\widehat{\mathcal{C}} = \mathcal{C}(\widehat{c})$, we let $\Sigma_{\widehat{\mathcal{C}}} = [\sigma_{\widehat{\mathcal{C}}}(x_i, x_j)]_{i,j=1}^{n \times n}$ be the similarity matrix associated with $\widehat{\mathcal{C}}$, i.e., $\sigma_{\widehat{\mathcal{C}}}(x_i, x_j) = 1$ if $x_i$ and $x_j$ belong to the same cluster, and $-1$ otherwise. Then the *Hamming* distance $d_H(\Sigma, \widehat{\mathcal{C}})$ simply counts the number of (ordered) pairs $(x_i, x_j)$ having inconsistent sign:

$$d_H(\Sigma, \widehat{\mathcal{C}}) = \left| \{ (x_i, x_j) \in L^2 \ : \ \sigma(x_i, x_j) \neq \sigma_{\widehat{\mathcal{C}}}(x_i, x_j) \} \right| \ .$$

The same definition applies in particular to the case when $\Sigma$ itself represents a clustering over $L$. The quantity $d_H$, sometimes called correlation clustering distance, is closely related to the Rand index [26] – see, e.g., [23]. Figure 2 contains illustrative examples.

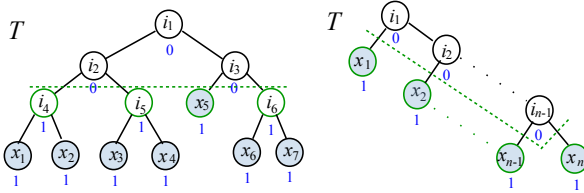

**Figure 3: Left:** The dotted green cut $c^*$ can be described by the set of values of $\{y(i), i \in V\}$, below each node. In this tree, in order to query, say, node $i_2$, it suffices to query any of the four pairs $(x_1, x_3)$, $(x_1, x_4)$, $(x_2, x_3)$, or $(x_2, x_4)$. The baseline queries $i_1$ through $i_6$ in a breadth-first manner, and then stops having identified $c^*$.

**Right:** This graph has $N(T) = n$. On the depicted cut, the baseline has to query all $n-1$ internal nodes.

**Prior distribution.** Recall cut $c^*$ defined in the noiseless realizable setting and its associated $\Sigma^*$. Depending on the specific learning model we consider (see below), the algorithm may have access to a *prior* distribution $\mathbb{P}(\cdot)$ over $c^*$, parametrized as follows. For $i \in V$, let $\boldsymbol{p}(i)$ be the conditional probability that $i$ is below $c^*$ given that all $i$'s ancestors are above. If we denote by $\mathcal{AB}(c^*) \subseteq V$ the nodes of $T$ which are above $c^*$, and by $\mathcal{LB}(c^*) \subseteq V$ those on the lower boundary of $c^*$, we can write

$$\mathbb{P}(c^*) = \left( \prod_{i \in \mathcal{AB}(c^*)} (1 - \boldsymbol{p}(i)) \right) \cdot \left( \prod_{j \in \mathcal{LB}(c^*)} \boldsymbol{p}(j) \right) , \tag{1}$$

where $\boldsymbol{p}(i) = 1$ if $i \in L$. In particular, setting $\boldsymbol{p}(i) = 1/N(i) \; \forall i$ yields the *uniform* prior $\mathbb{P}(c^*) = 1/N(T)$ for all $c^*$ realized by $T$. See Figure 1 (right) for an illustration. A canonical example of a non-uniform prior is one that favors cuts close to the root, which are thereby inducing clusterings having few clusters. These can be obtained, e.g., by setting $\boldsymbol{p}(i) = \alpha$, for some constant $\alpha \in (0, 1)$.

**Learning models.** We consider two learning settings. The first setting (Section 3) is an active learning setting under a noisy realizability assumption with prior information. Let $\mathcal{C}^* = \mathcal{C}(c^*)$ be the ground truth clustering induced by cut $c^*$ before noise is added. Here, for a given prior $\mathbb{P}(c^*)$, the goal of learning is to identify $\mathcal{C}^*$ either exactly (when $\lambda = 0$) or approximately (when $\lambda > 0$), while bounding the expected number of queries $(x_{i_t}, x_{j_t})$ made to the ground-truth matrix $\Sigma$, the expectation being over the noise, and possibly over $\mathbb{P}(c^*)$. In particular, if $\widehat{\mathcal{C}}$ is the clustering produced by the algorithm after it stops, we would like to prove upper bounds on $\mathbb{E}[d_H(\Sigma^*, \widehat{\mathcal{C}})]$, as related to the number of active learning rounds, as well as to the properties of the prior distribution. The second setting (Section 4) is a selective sampling setting where the pairs $(x_{i_t}, x_{j_t})$ are drawn i.i.d. according to an arbitrary and unknown distribution $\mathcal{D}$ over the $n^2$ entries of $\Sigma$, and the algorithm at every round can choose whether or not to query the label. After a given number of rounds the algorithm is stopped, and the goal is the typical goal of agnostic learning: no prior distribution over cuts is available anymore, and we would like to bound with high probability over the sample $(x_{i_1}, x_{j_1}), (x_{i_2}, x_{j_2}), \dots$ the so-called *excess risk* of the clustering $\widehat{\mathcal{C}}$ produced by $A$, i.e., the difference

$$\mathbb{P}_{(x_i, x_j) \sim \mathcal{D}} \left( \sigma(x_i, x_j) \neq \sigma_{\widehat{\mathcal{C}}}(x_i, x_j) \right) - \min_c \mathbb{P}_{(x_i, x_j) \sim \mathcal{D}} \left( \sigma(x_i, x_j) \neq \sigma_{\mathcal{C}(c)}(x_i, x_j) \right) , \tag{2}$$

the minimum being over all possible cuts $c$ realized by $T$. Notice that when $\mathcal{D}$ is uniform the excess risk reduces to $\frac{1}{n^2} \left( d_H(\Sigma, \widehat{\mathcal{C}}) - \min_c d_H(\Sigma, \mathcal{C}(c)) \right)$. At the same time, we would like to bound with high probability the total number of labels the algorithm has queried.

## 3 Active learning in the realizable case

As a warm up, we start by considering the case where $\lambda = 0$ (no noise). The underlying cut $c^*$ can be conveniently described by assigning to each node $i$ of $T$ a binary value $y(i) = 0$ if $i$ is above $c^*$, and $y(i) = 1$ if $i$ is below. Then we can think of an active learning algorithm as querying *nodes*, instead of querying pairs of leaves. A query to node $i \in V$ can be implemented by querying *any* pair $(x_{i_\ell}, x_{i_r}) \in L(\text{left}(i)) \times L(\text{right}(i))$. When doing so, we actually receive $y(i)$, since for any such $(x_{i_\ell}, x_{i_r})$, we clearly have $y(i) = \sigma^*(x_{i_\ell}, x_{i_r})$. An obvious baseline is then to perform a kind of breadth-first search in the tree: We start by querying the root $r$, and observe $y(r)$; if $y(r) = 1$ we stop and output clustering $\widehat{\mathcal{C}} = \{L\}$; otherwise, we go down by querying both $\text{left}(r)$ and $\text{right}(r)$, and then proceed recursively. It is not hard to show that this simple algorithm will make at most $2K - 1$ queries, with an overall running time of $O(K)$, where $K$ is the number of clusters of $\mathcal{C}(c^*)$. See Figure 3 for an illustration. If we know beforehand that $K$ is very small, then this baseline is a tough competitor. Yet, this is not the best we can do in general. Consider, for instance, the line graph in Figure 3 (right), where $c^*$ has $K = n$.

Ideally, for a given prior $\mathbb{P}(\cdot)$, we would like to obtain a query complexity of the form $\log(1/\mathbb{P}(c^*))$, holding in the worst-case for all underlying $c^*$. As we shall see momentarily, this is easily obtained when $\mathbb{P}(\cdot)$ is uniform. We first describe a version space algorithm (One Third Splitting, OTS) that admits a fast implementation, and whose number of queries in the worst-case is $\mathcal{O}(\log N(T))$. This will in turn pave the way for our second algorithm, Weighted Dichotomic Path (WDP). WDP leverages $\mathbb{P}(\cdot)$, but its theoretical guarantees only hold *in expectation* over $\mathbb{P}(c^*)$. WDP will then be extended to the persistent noisy setting through its variant Noisy Weighted Dichotomic Path (N-WDP).

We need a few ancillary definitions. First of all note that, in the noiseless setting, we have a clear hierarchical structure on the labels $y(i)$ of the internal nodes of $T$: Whenever a query reveals a label $y(i) = 0$, we know that all $i$'s ancestors will have label 0. On the other hand, if we observe $y(i) = 1$ we know that all internal nodes of subtree $T(i)$ have label 1. Hence, disclosing the label of some node indirectly entails disclosing the labels of either its ancestors or its descendants. Given $T$, a bottom-up path is any path connecting a node with one of its ancestors in $T$. In particular, we call a *backbone* path any bottom up path having maximal length. Given $i \in V$, we denote by $S_t(i)$ the *version space* at time $t$ associated with $T(i)$, i.e., the set of all cuts of $T(i)$ that are consistent with the labels revealed so far. For any node $j \neq i$, $S_t(i)$ splits into $S_t^{y(j)=0}(i)$ and $S_t^{y(j)=1}(i)$, the subsets of $S_t(i)$ obtained by imposing a further constraint on $y(j)$.

OTS **(One Third Splitting):** For all $i \in V$, OTS maintains over time the value $|S_t(i)|$, i.e., the size of $S_t(i)$, along with the forest $F$ made up of all maximal subtrees $T'$ of $T$ such that $|V(T')| > 1$ and for which none of their node labels have been revealed so far. OTS initializes $F$ to contain $T$ only, and maintains $F$ updated over time, by picking any backbone of any subtree $T' \in F$, and visiting it in a bottom-up manner. See the details in Appendix B.1. The following theorem (proof in Appendix B.1) crucially relies on the fact that $\pi$ is a backbone path of $T'$, rather than an arbitrary path.

**Theorem 1** *On a tree $T$ with $n$ leaves, height $h$, and number of cuts $N$, OTS finds $c^*$ by making $\mathcal{O}(\log N)$ queries. Moreover, an ad hoc data-structure exists that makes the overall running time $\mathcal{O}(n + h \log N)$ and the space complexity $\mathcal{O}(n)$.*

Hence, Theorem 1 ensures that, for all $c^*$, a time-efficient active learning algorithm exists whose number of queries is of the form $\log(1/\mathbb{P}(c^*))$, provided $\mathbb{P}(c^*) = 1/N(T)$ for all $c^*$. This query bound is fully in line with well-known results on splittable version spaces [12, 25, 24], so we cannot make claims of originality. Yet, what is relevant here is that this splitting can be done *very efficiently*.

We complement the above result with a *lower* bound holding in expectation over prior distributions on $c^*$. This lower bound depends in a detailed way on the structure of $T$. Given tree $T$, with set of leaves $L$, and cut $c^*$, recall the definitions of $\mathcal{AB}(c^*)$ and $\mathcal{LB}(c^*)$ we gave in Section 2. Let $T'_{c^*}$ be the subtree of $T$ whose nodes are $(\mathcal{AB}(c^*) \cup \mathcal{LB}(c^*)) \setminus L$, and then let $\widetilde{K}(T, c^*) = |L(T'_{c^*})|$ be the number of its leaves. For instance, in Figure 3 (left), $T'_{c^*}$ is made up of the six nodes $i_1, \ldots, i_6$, so that $\widetilde{K}(T, c^*) = 3$, while in Figure 3 (right), $T'_{c^*}$ has nodes $i_1, \ldots, i_{n-1}$, hence $\widetilde{K}(T, c^*) = 1$. Notice that we always have $\widetilde{K}(T, c^*) \leq K$, but for many trees $T$, $\widetilde{K}(T, c^*)$ may be much smaller than $K$. A striking example is again provided by the cut in Figure 3 (right), where $\widetilde{K}(T, c^*) = 1$, but $K = n$. It is also helpful to introduce $L_s(T)$, the set of all pairs of *sibling leaves* in $T$. For instance, in the tree of Figure 3, we have $|L_s(T)| = 3$. One can easily verify that, for all $T$ we have

$$\max_{c^*} \widetilde{K}(T, c^*) = |L_s(T)| \leq \log_2 N(T) .$$

We now show that there always exist families of prior distributions $\mathbb{P}(\cdot)$ such that the expected number of queries needed to find $c^*$ is $\Omega(\mathbb{E}[\widetilde{K}(T, c^*)])$. The quantity $\mathbb{E}[\widetilde{K}(T, c^*)]$ is our notion of *average (query) complexity*. Since the lower bound holds in expectation, it also holds in the worst case. The proof can be found in Appendix B.2.

**Theorem 2** *In the noiseless realizable setting, for any tree $T$, any positive integer $B \leq |L_s(T)|$, and any (possibly randomized) active learning algorithm $A$, there exists a prior distribution $\mathbb{P}(\cdot)$ over $c^*$ such that the* expected *(over $\mathbb{P}(\cdot)$ and $A$'s internal randomization) number of queries $A$ has to make in order to recover $c^*$ is lower bounded by $B/2$, while $B \leq \mathbb{E}[\widetilde{K}(T, c^*)] \leq 2B$, the latter expectation being over $\mathbb{P}(\cdot)$.*

Next, we describe an algorithm that, unlike OTS, is indeed able to take advantage of the prior distribution, but it does so at the price of bounding the number of queries only *in expectation*.

WDP **(Weighted Dichotomic Path)**: Recall prior distibution (1), collectively encoded through the values $\{\boldsymbol{p}(i), i \in V\}$. As for OTS, we denote by $F$ the forest made up of all maximal subtrees $T'$

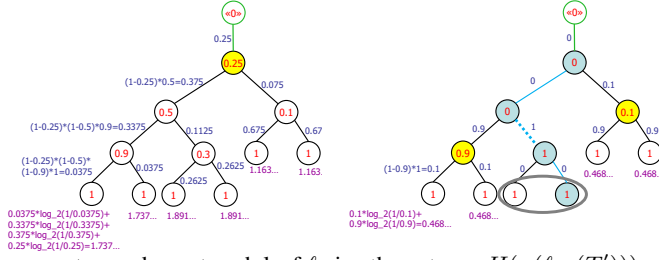

**Figure 4:** An example of input tree $T$ before (left) and after (right) the first binary search of WDP. The green node is a dummy super-root. The nodes in yellow are the roots of the subtrees currently included in forest $F$. The numbers in red within each node $i$ indicate the probabilities $\boldsymbol{p}(i)$, while the $q(i)$ values are in blue, and viewed here as associated with edges $(\mathrm{par}(i), i)$. The magenta numbers at each leaf $\ell$ give the entropy $H(\pi(\ell, r(T')))$, where $r(T')$ is the root of the subtree in $F$ that contains both $\ell$ and $r(T')$. **Left:** The input tree $T$ at time $t = 0$. No labels are revealed, and no clusters of $\mathcal{C}(c^*)$ are found. **Right:** Tree $T$ after a full binary search has been performed on the depicted light blue path. Before this binary search, that path connected a leaf of a subtree in $F$ to its root (in this case, $F$ contains only $T$). The selected path is the one maximazing entropy within the forest/tree on the left. The dashed line indicates the edge of $c^*$ found by the binary search. The red, blue and magenta numbers are updated accordingly to the result of the binary search. The leaves enclosed in the grey ellipse are now known to form a cluster of $\mathcal{C}(c^*)$.

of $T$ such that $|V(T')| > 1$ and for which none of their node labels have so far been revealed. $F$ is updated over time, and initially contains only $T$. We denote by $\pi(u, v)$ a bottom-up path in $T$ having as terminal nodes $u$ and $v$ (hence $v$ is an ancestor of $u$ in $T$). For a given cut $c^*$, and associated labels $\{y(i), i \in V\}$, any tree $T' \in F$, and any node $i \in V(T')$, we define[3]

$$q(i) = \mathbb{P}(y(i) = 1 \wedge y(\mathrm{par}(i)) = 0) = \boldsymbol{p}(i) \cdot \prod_{j \in \pi(\mathrm{par}(i), r(T'))} (1 - \boldsymbol{p}(j)). \tag{3}$$

We then associate with any backbone path of the form $\pi(\ell, r(T'))$, where $\ell \in L(T')$, an entropy $H(\pi(\ell, r(T'))) = -\sum_{i \in \pi(\ell, r(T'))} q(i) \log_2 q(i)$. Notice that at the beginning we have $\sum_{i \in \pi(\ell, r(T))} q(i) = 1$ for all $\ell \in L$. This invariant will be maintained on all subtrees $T'$. The prior probabilities $\boldsymbol{p}(i)$ will evolve during the algorithm's functioning into posterior probabilities based on the information revealed by the labels. Accordingly, also the related values $q(i)$ w.r.t. which the entropy $H(\cdot)$ is calculated will change over time.

Due to space limitations, WDP's pseudocode is given in Appendix B.3, but we have included an example of its execution in Figure 4. At each round, WDP finds the path whose entropy is maximized over all bottom-up paths $\pi(\ell, r')$, with $\ell \in L$ and $r' = r(T')$, where $T'$ is the subtree in $F$ containing $\ell$. WDP performs a binary search on such $\pi(\ell, r')$ to find the edge of $T'$ which is cut by $c^*$, taking into account the current values of $q(i)$ over that path. Once a binary search terminates, WDP updates $F$ and the probabilities $\boldsymbol{p}(i)$ at all nodes $i$ in the subtrees of $F$. See Figure 4 for an example. Notice that the $\boldsymbol{p}(i)$ on the selected path become either 0 (if above the edge cut by $c^*$) or 1 (if below). In turn, this causes updates on all probabilities $q(i)$. WDP continues with the next binary search on the next path with maximum entropy at the current stage, discovering another edge cut by $c^*$, and so on, until $F$ becomes empty. Denote by $\mathcal{P}_{>0}$ the set of all priors $\mathbb{P}(\cdot)$ such that for all cuts $c$ of $T$ we have $\mathbb{P}(c) > 0$. The proof of the following theorem is given in Appendix B.3.

**Theorem 3** *In the noiseless realizable setting, for any tree $T$ of height $h$, any prior distribution $\mathbb{P}(\cdot)$ over $c^*$, such that $\mathbb{P}(\cdot) \in \mathcal{P}_{>0}$, the expected number of queries made by* WDP *to find $c^*$ is $\mathcal{O}\left(\mathbb{E}\left[\widetilde{K}(T, c^*)\right] \log h\right)$, the expectations being over $\mathbb{P}(\cdot)$.*

For instance, in the line graph of Figure 3 (right), the expected number of queries is $\mathcal{O}(\log n)$ for any prior $\mathbb{P}(\cdot)$, while if $T$ is a complete binary tree with $n$ leaves, and we know that $\mathcal{C}(c^*)$ has $\mathcal{O}(K)$ clusters, we can set $\boldsymbol{p}(i)$ in (1) as $\boldsymbol{p}(i) = 1/\log K$, which would guarantee $\mathbb{E}[\widetilde{K}(T, c^*)] = \mathcal{O}(K)$, and a bound on the expected number of queries of the form $\mathcal{O}(K \log \log n)$. By comparison, observe that the results in [12, 15, 27] would give a query complexity which is at best $\mathcal{O}(K \log^2 n)$, while those in [25, 24] yield at best $\mathcal{O}(K \log n)$. In addition, we show below (Remark 1) that our algorithm has very compelling running time guarantees.

It is often the case that a linkage function generating $T$ also tags each internal node $i$ with a *coherence level* $\alpha_i$ of $T(i)$, which is typically increasing as we move downwards from root to leaves. A common

situation in hierarchical clustering is then to figure out the "right" level of granularity of the flat clustering we search for by defining parallel *bands* of nodes of similar coherence where $c^*$ is possibly located. For such cases, a slightly more involved guarantee for WDP is contained in Theorem 6 in Appendix B.3, where the query complexity depends in a more detailed way on the interplay between $T$ and the prior $\mathbb{P}(\cdot)$. In the above example, if we have $b$-many edge-disjoint bands, Theorem 6 replaces factor $\log h$ of Theorem 3 by $\log b$.

N-WDP (**Noisy Weighted Dichotomic Path**): This is a robust variant of WDP that copes with persistent noise. Whenever a label $y(i)$ is requested, N-WDP determines its value by a majority vote over randomly selected pairs from $L(\text{left}(i)) \times L(\text{right}(i))$. Due to space limitations, all details are contained in Appendix B.4. The next theorem quantifies N-WDP's performance in terms of a tradeoff between the expected number of queries and the distance to the noiseless ground-truth matrix $\Sigma^*$.

**Theorem 4** *In the noisy realizable setting, given any input tree $T$ of height $h$, any cut $c^* \sim \mathbb{P}(\cdot) \in \mathcal{P}_{>0}$, and any $\delta \in (0, 1/2)$, N-WDP outputs with probability $\geq 1 - \delta$ (over the noise in the labels) a clustering $\widehat{\mathcal{C}}$ such that $\frac{1}{n^2} d_H(\Sigma^*, \widehat{\mathcal{C}}) = \mathcal{O}\left(\frac{1}{n} \frac{(\log(n/\delta))^{3/2}}{(1-2\lambda)^3}\right)$ by asking $\mathcal{O}\left(\frac{\log(n/\delta)}{(1-2\lambda)^2} \mathbb{E}\widetilde{K}(T, c^*) \log h\right)$ queries in expectation (over $\mathbb{P}(\cdot)$).*

**Remark 1** *Compared to the query bound in Theorem 3, the one in Theorem 4 adds a factor due to noise. The very same extra factor is contained in the bound of [22]. Regarding the running time of WDP, the version we have described can be naively implemented to run in $\mathcal{O}(n \, \mathbb{E}\widetilde{K}(T, c^*))$ expected time overall. A more time-efficient variant of WDP exists for which Theorem 3 and Theorem 6 still hold, that requires $\mathcal{O}(n + h \, \mathbb{E}\widetilde{K}(T, c^*))$ expected time. Likewise, an efficient variant of N-WDP exists for which Theorem 4 holds, that takes $\mathcal{O}\left(n + \left(h + \frac{\log^2 n}{(1-2\lambda)^2}\right) \mathbb{E}\widetilde{K}(T, c^*)\right)$ expected time.*

## 4 Selective sampling in the non-realizable case

In the non-realizable case, we adapt to our clustering scenario the importance-weighted algorithm in [6]. The algorithm is a selective sampler that proceeds in a sequence of rounds $t = 1, 2, \ldots$. In round $t$ a pair $(x_{i_t}, x_{j_t})$ is drawn at random from distribution $\mathcal{D}$ over the entries of a given ground truth matrix $\Sigma$, and the algorithm produces in response a probability value $p_t = p_t(x_{i_t}, x_{j_t})$. A Bernoulli variable $Q_t \in \{0, 1\}$ is then generated with $\mathbb{P}(Q_t = 1) = p_t$, and if $Q_t = 1$ the label $\sigma_t = \sigma(x_{i_t}, x_{j_t})$ is queried, and the algorithm updates its internal state; otherwise, we skip to the next round. The way $p_t$ is generated is described as follows. Given tree $T$, the algorithm maintains at each round $t$ an importance-weighted empirical risk minimizer cut $\hat{c}_t$, defined as $\hat{c}_t = \text{argmin}_c \, \text{err}_{t-1}(\mathcal{C}(c))$, where the "argmin" is over all cuts $c$ realized by $T$, and $\text{err}_{t-1}(\mathcal{C}) = \frac{1}{t-1} \sum_{s=1}^{t-1} \frac{Q_s}{p_s} \{\sigma_{\mathcal{C}}(x_{i_s}, x_{j_s}) \neq \sigma_s\}$, being $\{\cdot\}$ the indicator function of the predicate at argument. This is paired up with a *perturbed empirical risk minimizer*

$$\hat{c}_t' = \underset{c \,:\, \sigma_{\mathcal{C}(c)}(x_{i_t}, x_{j_t}) \neq \sigma_{\mathcal{C}(\hat{c}_t)}(x_{i_t}, x_{j_t})}{\text{argmin}} \text{err}_{t-1}(\mathcal{C}(c)) \,,$$

the "argmin" being over all cuts $c$ realized by $T$ that disagree with $\hat{c}_t$ on the current pair $(x_{i_t}, x_{j_t})$. The value of $p_t$ is a function of $d_t = \text{err}_{t-1}(\mathcal{C}(\hat{c}_t')) - \text{err}_{t-1}(\mathcal{C}(\hat{c}_t))$, of the form

$$p_t = \min\left\{1, \mathcal{O}\left(1/d_t^2 + 1/d_t\right) \log((N(T)/\delta)\log t)/t\right\} \,, \tag{4}$$

where $N(T)$ is the total number of cuts realized by $T$ (i.e., the size of our comparison class), and $\delta$ is the desired confidence parameter. Once stopped, say in round $t_0$, the algorithm gives in output cut $\hat{c}_{t_0+1}$, and the associated clustering $\mathcal{C}(\hat{c}_{t_0+1})$. Let us call the resulting algorithm NR (Non-Realizable).

Despite $N(T)$ can be exponential in $n$, there are very efficient ways of computing $\hat{c}_t$, $\hat{c}_t'$, and hence $p_t$ at each round. In particular, an ad hoc procedure exists that incrementally computes these quantities by leveraging the sequential nature of NR. For a given $T$, and constant $K \geq 1$, consider the class $\mathbb{C}(T, K)$ of cuts inducing clusterings with at most $K$ clusters. Set $R^* = R^*(T, \mathcal{D}) = \min_{c \in \mathbb{C}(T, K)} \mathbb{P}_{(x_i, x_j) \sim \mathcal{D}}\left(\sigma(x_i, x_j) \neq \sigma_{\mathcal{C}(c)}(x_i, x_j)\right)$, and $B_\delta(K, n) = K \log n + \log(1/\delta)$. The following theorem is an adaptation of a result in [6]. See Appendix C.1 for a proof.

**Theorem 5** *Let $T$ have $n$ leaves and height $h$. Given confidence parameter $\delta$, for any $t \geq 1$, with probability at least $1 - \delta$, the excess risk (2) achieved by the clustering $\mathcal{C}(\hat{c}_{t+1})$ computed by NR w.r.t. the best cut in class $\mathbb{C}(T, K)$ is bounded by $\mathcal{O}\left(\sqrt{\frac{B_\delta(K, n) \log t}{t}} + \frac{B_\delta(K, n) \log t}{t}\right)$, while the (expected)*

| Tree | Avg depth | Std. dev | BEST's error | BEST's $K$ |
|------|-----------|----------|--------------|------------|
| SING | 2950 | 1413.6 | 8.26% | 4679 |
| MED | 186.4 | 41.8 | 8,51% | 1603 |
| COMP | 17.1 | 3.3 | 8.81% | 557 |

**Table 1:** Statistics of the trees used in our experiments. These trees result from applying the linkage functions SING, COMP, and MED to the MNIST dataset (first 10000 samples). Each tree has the same set of $n = 10000$ leaves. "Avg depth" is the average depth of the leaves in the tree, "Std. dev" is its standard deviation. For reference, we report the performance of BEST (i.e., the minimizer of $d_H$ over all possible cuts realized by the trees), along with the associated number of clusters $K$.

number of labels $\sum_{s=1}^{t} p_s$ is bounded by $\mathcal{O}\left(\theta\left(R^* t + \sqrt{t\,B_\delta(K,n)\log t} + B_\delta(K,n)\log^3 t\right)\right)$, where $\theta = \theta(\mathbb{C}(T,K), \mathcal{D})$ is the disagreement coefficient of $\mathbb{C}(T,K)$ w.r.t. distribution $\mathcal{D}$. In particular, when $\mathcal{D}$ is uniform we have $\theta \leq K$. Moreover, there exists a fast implementation of NR whose expected running time per round is $\mathbb{E}_{(x_i,x_j)\sim\mathcal{D}}[\mathrm{de}(\mathrm{lca}(x_i,x_j))] \leq h$, where $\mathrm{de}(\mathrm{lca}(x_i,x_j))$ is the depth in $T$ of the lowest common ancestor of $x_i$ and $x_j$.

## 5 Preliminary experiments

The goal of these experiments was to contrast active learning methods originating from the persistent noisy setting (specifically, N-WDP) to those originating from the non-realizable setting (specifically, NR). The comparison is carried out on the hierarchies produced by standard HC methods operating on the first $n = 10000$ datapoints in the well-known MNIST dataset from `http://yann.lecun.com/exdb/mnist/`, yielding a sample space of $10^8$ pairs. We used Euclidean distance combined with the single linkage (SING), median linkage (MED), and complete linkage (COMP) functions. The $n \times n$ ground-truth matrix $\Sigma$ is provided by the 10 class labels of MNIST.

We compared N-WDP with uniform prior and NR to two baselines: passive learning based on empirical risk minimization (ERM), and the active learning baseline performing breadth-first search from the root (BF, Section 3) made robust to noise as in N-WDP. For reference, we also computed for each of the three hierarchies the performance of the best cut in hindsight (BEST) on the *entire* matrix $\Sigma$. That is essentially the best one can hope for in each of the three cases. All algorithms except ERM are randomized and have a single parameter to tune. We let such parameters vary across suitable ranges and, for each algorithm, picked the best performing value on a validation set of 500 labeled pairs.

In Table 1, we have collected relevant statistics about the three hierarchies. In particular, the single linkage tree turned out to be very deep, while the complete linkage one is quite balanced. We evaluated test set accuracy vs. number of queries after parameter tuning, excluding these 500 pairs. For N-WDP, once a target number of queries was reached, we computed as current output the maximum-a-posteriori cut. In order to reduce variance, we repeated each experiment 10 times.

The details of our empirical comparison are contained in Appendix C.3. Though our experiments are quite preliminary, some trends can be readily spotted (see Table 2 in Appendix C.3): i. N-WDP significantly outperforms NR. E.g., in COMP at 250 queries, the test set accuracy of N-WDP is at 9.52%, while NR is at 10.1%. A similar performance gap at low number of queries one can observe in SING and MED. This trend was expected: NR is very conservative, as it has been designed to work under more general conditions than N-WDP. We conjecture that, whenever the specific task at hand allows one to make an aggressive noise-free algorithm (like WDP) robust to persistent noise (like N-WDP), this outcome is quite likely to occur. ii. BF is competitive only when BEST has few clusters. iii. N-WDP clearly outperforms ERM, while the comparison between NR and ERM yields mixed results.

**Conclusions and ongoing activity.** Beyond presenting new algorithms and analyses for pairwise similarity-based active learning, our goal was to put different approaches to active learning on the same footing for comparison on real data. The initial evidence emerging from our experiments is that Active Learning algorithms based on persistent noise can in practice be more effective than those making the more general non-realizable assumption. Notice that the similarities of the pairs of items have been generated by the MNIST class labels, hence they have virtually nothing to do with the trees we generated, which in turn do not rely on those labels at all. These initial trends suggested by our experiments clearly need a more thorough investigation. We are currently using other datasets, of different nature and size. Further HC methods are also under consideration, like those based on $k$-means.

**Acknowledgments**

Fabio Vitale acknowledges support from the Google Focused Award "ALL4AI" and the ERC Starting Grant "DMAP 680153", awarded to the Department of Computer Science of Sapienza University.

## Footnotes

[1] In fact, the trees we can handle are more general than binary: we are making the binary assumption throughout for presentational convenience only.

[2] A strongly binary tree is a rooted binary tree for which the root is adjacent to either zero or two nodes, and all non-root nodes are adjacent to either one or three nodes.

[3] For definiteness, we set $y(\mathrm{par}(r)) = 0$, that is, we are treating the parent of $r(T)$ as a "dummy super–root" with labeled 0 since time $t = 0$. Thus, according to this definition, $q(r) = \boldsymbol{p}(r)$.

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
