[Supplementary Material · paper2.pdf]

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

[4] This bound is indeed tight for this strategy when the input is a full binary tree of height 3.

[5] Here, we are defining the entropy of a path $\pi$ as $- \sum_{v \in V(\pi)} q(v) \log_2 q(v)$, even for paths $\pi$ for which $\sum_{v \in V(\pi)} q(v) < 1$.

[6] Note that $a_t$ can always be found in constant time after a $\Theta(n)$ time preprocessing phase of $T$ – see [19].

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

# A  Missing material from Section 1

## A.1  Further related work

Further papers related to our work are those dealing with clustering with queries, e.g., [13, 2, 22, 4, 3]. In [13] the authors show that $\mathcal{O}(Kn)$ similarity queries are both necessary and sufficient to achieve exact reconstruction of an *arbitrary* clustering with $K$ clusters on $n$ items. This is generalized by [22] where persistent random noise is added. [2] assume the feedback is center-based with a margin condition on top. Because we are constrained to a clustering produced by cutting a given tree, the results in [13, 2, 22] are incomparable to ours, due to the different assumptions. In [4, 3]) the authors consider clusterings realized by a given comparison class (as we do here). Yet, the queries they are allowing are different from ours, hence their results are again incomparable to ours.

Further relevant papers include [29, 28, 14], though these papers are not readily comparable to ours, since their noise assumptions are slightly different, see, e.g., the uneven noise gap analysis in [14] (Theorem 1 and 2 and Propositions 3 and 5 therein). Moreover, being restricted to a clustering realized by $T$, we generally need less queries. For instance, combining our Theorems 2 and 3, our bound in the realizable case is between $\mathbb{E}[K]$ and $\mathbb{E}[K] \log h$, while in general it takes $n\mathbb{E}[K]$ queries to fully reconstruct the clustering [13]. Also recall the lower bound $n - K + \binom{K}{2}$ in [29] and [14].

# B  Missing material from Section 3

## B.1  One Third Splitting (OTS)

For all $i \in V$, OTS maintain over time the value $|S_t(i)|$, i.e., the size of $S_t(i)$, along with the forest $F$ made up of all maximal subtrees $T'$ of $T$ such that $|V(T')| > 1$ and for which none of their node labels have been revealed so far. Notice that we will not distinguish between labels $y(i)$ revealed directly by a query or indirectly by the hierarchical structure. By maximal here we mean that it is not possible to extend any such subtrees by adding a node of $V$ whose label has not already been revealed. OTS initializes $F$ (when no labels are revealed) to contain $T$ only, and maintains $F$ updated over time. Let subtree $T' \in F$ be arbitrarily chosen, and $\pi$ be any backbone path of $T'$. At time $t$, OTS visits $\pi$ in a bottom-up manner, and finds the *lowest* node $i_t^*$ in this path satisfying $\left|S_t^{y(i_t^*)=1}(r)\right| \leq 2\left|S_t^{y(i_t^*)=0}(r)\right|$, i.e., $\left|S_t^{y(i_t^*)=1}(r)\right| \leq \frac{2}{3}|S_t(r)|$, then query node $i_t^*$. We repeat the above procedure until $|S_t(r)| = 1$, i.e., until we find $c^*$.

The next lemma is key to showing the logarithmic number of queries made by OTS.

**Lemma 1** *With the notation introduced in Section 3, at each time $t$, the query $y(i_t^*)$ made by OTS splits the version space $S_t(r)$ in such a way that*[4]

$$\min\left\{\left|S_t^{y(i_t^*)=0}(r)\right|, \left|S_t^{y(i_t^*)=1}(r)\right|\right\} \geq \frac{|S_t(r)|}{3} .$$

*Proof.* At each time $t$, $|S_t(r)|$ is the product of the cardinality of $S_t(\widetilde{r})$ over all roots $\widetilde{r}$ of the trees currently contained in $F$. Let $\pi$ be a backbone of one such tree, say tree $T'$, with root $r'$. Since $T'$ is arbitrary, in order to prove the statement, it is sufficient to show that

$$\min\left\{\left|S_t^{y(i_t^*)=0}(r')\right|, \left|S_t^{y(i_t^*)=1}(r')\right|\right\} \geq \frac{|S_t(r')|}{3} .$$

Let $h(\pi)$ be the length of $\pi$, i.e., the number of its edges, and $\langle j_0, j_1, \ldots, j_{h(\pi)}\rangle$ be the sequence of its nodes, from bottom to top. For any $k < h(\pi)$, we denote by $j_k^{\mathrm{s}}$ the sibling of $j_k$ in $T'$ (hence, by this definition $j_k^{\mathrm{s}}$ does *not* belong to $\pi$). Now, observe that the number of possible labelings of $\pi$ is equal to $h(\pi) + 1$, that is, each labeling of $\pi$ corresponds to an integer $z \in \{0, 1, \ldots, h(\pi)\}$ such that $y(j_k) = 1$ for all $k \leq z$ and $y(j_k) = 0$ for all $z < k \leq h(\pi)$. Then, given any labeling of the nodes of $\pi$ (represented by the above $z$), we have

$$|S_t(r')| = \begin{cases} \prod_{k=z}^{h(\pi)-1} |S_t(j_k^{\mathrm{s}})| & \text{if } z < h(\pi) , \\ 1 & \text{if } z = h(\pi) . \end{cases}$$

**Figure 5:** A backbone path $\pi$ selected by OTS and the associated quantities needed to prove the main properties of the selected node $i_t^*$. Leaves are represented by squares, subtrees by triangles. For simplicity, in this picture $\pi$ is starting from the leftmost leaf of $T'$, but it can clearly be chosen to start from any of its deepest leaves. The sum of all terms on the right of each subtree equals $|S_t(r')|$. The cornerstone of the proof is that $i_t^*$ (and hence $z^*$) corresponds to the lowest among the $h(\pi) = 4$ horizontal lines depicted in this figure for which the sum of all products below the chosen line is at least half the sum of all the products above the line. Furthermore, the fact that $|S_t(j_0^s)| = 1$ guarantees that $\mathcal{S}_0 = \mathcal{S}_1$. Combined with the fact that $\mathcal{S}_z \geq \mathcal{S}_{z+1}$ for all $z \in \{0, \ldots, h(\pi) - 1\}$, this ensures that the abovementioned horizontal line always exists, and splits the sum of all $h(\pi) + 1$ terms into two parts such that the smaller one is at least $\frac{1}{3}$ of the total.

In fact, the disclosure of all labels of the nodes in $\pi$ when $z < h(\pi)$ would decompose $T'$ into $(h(\pi) - z)$-many subtrees whose labelings are independent of one another. For all $z \in \{0, 1, \ldots, h(\pi) - 1\}$, let us denote for brevity $\prod_{k=z}^{h(\pi)-1} |S_t(j_k^s)|$ by $\mathcal{S}_z$, and also denote for convenience $|S_t^{y(r')=1}(r')|$ by $\mathcal{S}_{h(\pi)}$. Notice that, by definition, $|S_t^{y(r')=1}(r')| = 1$, and corresponds to the special case $z = h(\pi)$. With this notation, it is now important to note that $i_t^*$ must be the parent of $j_{z^*}^s$, for some $z^* \in \{0, 1, \ldots, h(\pi) - 1\}$, and that $\left|S_t^{y(i_t^*)=0}(r')\right| = \mathcal{S}_{z^*}$. Thus, taking into account all possible $(h(\pi) + 1)$ labelings of $\pi$, the cardinality of $S_t(r')$ can be written as follows:

$$|S_t(r')| = \sum_{z=0}^{h(\pi)} \mathcal{S}_z .$$

At this point, by definition, we have:

(i) $\mathcal{S}_0 = \mathcal{S}_1$, as $|S_t(j_0^s)| = 1$, which in turn implies $\max_z \mathcal{S}_z \leq \frac{|S_t(r')|}{2}$, and

(ii) $\mathcal{S}_z \geq \mathcal{S}_{z+1}$ for all $z \in \{0, \ldots, h(\pi) - 1\}$ .

See Figure 5 for a pictorial illustration.

The proof is now concluded by contradiction. If our statement is false, then there must exist a value $z'$ such that $\mathcal{S}_{z'} > \frac{2}{3}|S_t(r')|$ and $\mathcal{S}_{z'+1} < \frac{1}{3}|S_t(r')|$. However, because the sequence $\langle \mathcal{S}_0, \mathcal{S}_1, \ldots \mathcal{S}_{h(\pi)}\rangle$ is monotonically decreasing and we have $\mathcal{S}_0 \leq \frac{|S_t(r')|}{2}$, implying $\mathcal{S}_0 \leq \sum_{z=1}^{h(\pi)} \mathcal{S}_z$, such value $z'$ cannot exist. Thus, it must exist $z$ such that

$$\frac{1}{3}|S_t(r')| \leq \mathcal{S}_z \leq \frac{2}{3}|S_t(r')| .$$

Let $z^*$ be the smallest $z$ satisfying the above inequalities. Note that $i_t^*$ is the parent of $j_{z^*}^s$, because of the bottom-up search on $\pi$ performed by OTS. Exploiting again the monotonicity of the sequence $\langle \mathcal{S}_0, \mathcal{S}_1, \ldots \mathcal{S}_{h(\pi)}\rangle$ and recalling that $\mathcal{S}_{z^*} = \prod_{k=z^*}^{h(\pi)-1} |S_t(j_k^s)|$, we conclude that

$$\frac{1}{3}|S_t(r')| \leq \left|S_t^{y(i_t^*)=0}(r')\right| \leq \frac{2}{3}|S_t(r')| .$$

Since $\left|S_t^{y(i_t^*)=1}(r')\right| + \left|S_t^{y(i_t^*)=0}(r')\right| = |S_t(r')|$, we must also have

$$\frac{1}{3}|S_t(r')| \leq \left|S_t^{y(i_t^*)=1}(r')\right| \leq \frac{2}{3}|S_t(r')|,$$

thereby concluding the proof. $\square$

From the above proof, one can see that it is indeed necessary that $\pi$ is a backbone path, since the proof hinges on the fact that $|S_t(j_0^{\mathrm{s}})| = 1$. In fact, if $|S_t(j_0^{\mathrm{s}})|$ is larger than $2\sum_{z=1}^{h(\pi)}|\mathcal{S}_z|$, that is larger than $\frac{2}{3}|S_t(r')|$ (which may happen if $\pi$ is not a backbone path), we would not have $\max_z \mathcal{S}_z \le \frac{|S_t(r')|}{2}$, hence $\mathcal{S}_z$ would not be guaranteed to be at least $\frac{2}{3}|S_t(r')|$ for all $z$.

**Proof of Theorem 1**

*Proof.* By Lemma 1, we immediately see that OTS finds $c^*$ through $\mathcal{O}(\log N)$ queries. This is because $|S_{t+1}(r)| \le \frac{2}{3}|S_t(r)|$ for all time steps $t$, implying by induction that the total number of queries is upper bounded by $\log_{3/2} N = \mathcal{O}(\log N)$.

We now sketch an implementation of OTS which requires $\mathcal{O}(n + h\log N)$ time and $\mathcal{O}(n)$ space.

In a preliminary phase, we compute in a bottom-up fashion the values $|S_0(i)|$ for all nodes $i \in V(T)$. This requires $\mathcal{O}(n)$. Thereafter, we perform a breath-first search on $T$, and each time we visit a leaf of $T$, we insert a pointer to it in a an array $A$ in a sequential way. Thus, the $j$-th record of $A$ will contain a reference to the $j$-th leaf found during this visit, which entails that the leaves referred by the pointers of $A$ are sorted in ascending order of depth.

We recall that in the noiseless setting, each time the label of a node is revealed and is equal to 1 (to 0), also the labels of its descendants (ancestors) are indirectly revealed, because they are known to be equal to 1 (to 0). The total time OTS takes for assigning all indirectly revealed labels is clearly $\mathcal{O}(n)$. Each time OTS needs to find a backbone of a tree in the current forest $F$, we look for the largest index $j$ for which the record $A[j]$ does not point to a leaf whose parent label has not been revealed yet. Observe that, at any time $t$, the deepest leaf $\ell \in L(T)$ satisfying this property must be the terminal node of a backbone path of a tree in $F$. Furthermore, the highest node of such backbone is either $r(T)$ or the lowest ancestor of $\ell$ whose label has not been revealed yet, and can therefore be found in $\mathcal{O}(h)$ time.

In order to accomplish this leaf search operation, we simply maintain over time an index that scans $A$ from $A[n]$ to $A[1]$, looking for a leaf satisfying the above property. The total time OTS uses for scanning $A$ is again linear in $n$. Finally, for each query, OTS traverses bottom-up a backbone $\pi$, exploiting the information previously stored to find $i_t^*$, and updates it after $y(i_t^*)$ is revealed. Note that only the information of the nodes in $\pi$ has to be updated. In fact, the disclosure of the label of any node $i \in V(T)$ cannot affect the values of $S_t(j)$ for all nodes $j \in V(T)$ that are *not* ancestors of $i$. Besides, we are free to disregard the descendants of $i$ since they will simply be indirectly labeled (by 1).

Overall, the total time required by this implementation of OTS is the sum of $\mathcal{O}(n)$ and $\mathcal{O}(h)$ times the total number of queries the algorithm makes, which results in the claimed $\mathcal{O}(n + h\log N)$ upper bound. The claim on the memory requirement immediately follows from the above description. $\square$

## B.2   Proof of the lower bound in Theorem 2

*Proof.* Let $T'$ be the subtree of $T$ constructed by visiting $T$ from its root (for instance by a breadth-first or a depth-first visit), and such that $|L(T')| = B$. Note that the construction of $T'$ satisfying this constraint is always possible because the maximum cardinality of $L(T')$ is equal to $|L_{\mathrm{s}}(T)|$ (which is also equal to $\max_{c^*} \widetilde{K}(T, c^*)$). For each leaf $\ell \in L(T')$, consider all cuts $c^*$ that can be generated by cutting either the edge connecting $\ell$ with its parent or the two edges connecting $\ell$ with its children. The total number of such cuts is $2^{|L(T')|} = 2^B$. We set the prior $\mathbb{P}(\cdot)$ to be uniform over these $2^B$-many cuts. Hence, for each leaf $\ell \in L(T')$, the probability (w.r.t. $\mathbb{P}(\cdot)$) that $c^*$ cuts the edge connecting $\ell$ with its parent is 1/2, and so is the probability that $c^*$ cuts the two edges connecting $\ell$ with its children.

Now, observe that, by construction, we have $B \le \widetilde{K}(T, c^*) \le 2B$ for *all* such cuts $c^*$ and, as a consequence, $B \le \mathbb{E}[\widetilde{K}(T, c^*)] \le 2B$, the expectation being over $\mathbb{P}(\cdot)$. Since for each leaf of $T'$ any (possibly randomized) active learning algorithm $A$ has to make $\frac{1}{2}$ mistake in expectation (over $\mathbb{P}(\cdot)$ and its internal randomization), we conclude that $B/2$ queries are always necessary to find $c^*$, as claimed. $\square$

**Algorithm 1:** WDP (Weighted Dichotomic Path)

▷ **INPUT** : $T$, $\{\boldsymbol{p}(i), i \in V\}$.
▷ **OUTPUT**: $\widehat{\mathcal{C}} = \mathcal{C}^*$.
**Init:** • $\widehat{\mathcal{C}} \leftarrow \emptyset$; /* $\widehat{\mathcal{C}}$ contains all the clusters of $\mathcal{C}(c^*)$ found so far */
   • $F \leftarrow \{T\}$; /* Forest of maximal subtrees $T'$ of $T$ */
   • $y(\mathrm{par}(r)) \leftarrow 0$; /* Dummy node $\mathrm{par}(r)$ */
   • **for** $i \in V$ **do**  $q(i) \leftarrow \boldsymbol{p}(i) \cdot \prod_{j \in \pi(\mathrm{par}(i), r(T))}(1 - \boldsymbol{p}(j))$;

/*  -- Path with maximum entropy --  */
**while** $F \neq \emptyset$ **do**
  Let $L(F)$ be the set of all leaves of $T$ belonging to the subtrees in $F$.
  Let $R(F)$ be the set of all roots of the subtrees in $F$.
  $\pi(\ell, r') \leftarrow \arg\max_{\ell \in L(F), r' \in R(F)} H(\pi(\ell, r'))$;
  $\mathcal{T} = \{T' \in F : \ell, r' \in V(T')\}$;

  /*  -- Binary search on path $\pi(\ell, r')$ --  */
  $u \leftarrow \ell$;   $v \leftarrow r'$;
  **while** $u \neq v$ **do**
    Let $\langle i_0 = u, i_1, \ldots, i_{h-1}, i_h = v \rangle$ be the sequence of nodes lying on $\pi(u, v)$ in descending order of depth.
    $Q \leftarrow \sum_{k \in \{0, 1, \ldots, h-1\}} q(i_k)$;
    $k^* = \arg\min_{k \in \{0, 1, \ldots, h-1\}} \left| \frac{Q}{2} - \sum_{j \in \{i_0, \ldots, i_k\}} q(j) \right|$;
    $i^* \leftarrow \mathrm{par}(i_{k^*})$;
    Query $y(i^*)$;
    **if** $y(i^*) = 0$ **then**
      $v \leftarrow i_{k^*}$;
    **else**
      $u \leftarrow i^*$;

  Set $\boldsymbol{p}(i) = y(i) = 1$ for all descendants $i$ of $u$;
  Set $\boldsymbol{p}(i) = y(i) = 0$ for all ancestors $i \neq u$ of $u$;
  $q(u) \leftarrow 1$;     Set $q(i) = 0$ for all descendants and ancestors $i \neq u$ of $u$;
  Update $\boldsymbol{p}(i)$ and $q(i)$ for all descendants of all $i \in V(\pi(\mathrm{par}(u), r'))$ such that $i \neq r'$;
  $\widehat{\mathcal{C}} \leftarrow \widehat{\mathcal{C}} \cup \{L(u)\}$; /* $L(u)$ is a cluster of $\mathcal{C}^*$ */

  /*  -- Update F --  */
  $F \leftarrow F \setminus \mathcal{T}$; /* Remove from F the subtree containing $\pi(\ell, r')$ */
  $j \leftarrow \mathrm{par}(u)$; /* Lowest node in $\pi(\ell, \mathrm{par}(r'))$ with label known to be 0 */
  **while** $j \in V(\pi(\ell, r'))$ **do**
    Let $j_c$ be the child of $j$ that is not in $\pi(\ell, r')$.
    **if** $j_c \in L$ **then**
      $\widehat{\mathcal{C}} \leftarrow \widehat{\mathcal{C}} \cup \{j_c\}$; /* Add to $\widehat{\mathcal{C}}$ a singleton cluster */
    **else**
      $F \leftarrow F \cup \{T(j_c)\}$; /* $j_c$ is the root of a subtree that will be processed later */
      Update $q(i)$ for all $i \in V(T(j_c))$;
    $j \leftarrow \mathrm{par}(j)$; /* j is a node whose label is known to be 0 */

**return** $\widehat{\mathcal{C}}$ .

---

### B.3  Weighted Dichotomic Path (WDP)

In Algorithm 1 we give the pseudocode of WDP. At each round, WDP finds the path whose entropy is maximized over all bottom-up paths $\pi(\ell, r')$, with $\ell \in L$ and $r' = r(T')$, where $T'$ is the subtree in $F$ containing $\ell$. Ties are broken arbitrarily. WDP performs a binary search on such $\pi(\ell, r')$ to find the edge of $T'$ which is cut by $c^*$, taking into account the current values of $q(i)$ over that path.

Specifically, let $\langle i_0 = \ell, i_1, \ldots, i_{h-1}, i_h = r' \rangle$ be the sequence of nodes in $\pi(\ell, r')$ in descending order of depth. WDP finds an index $k^*$ that corresponds to the middle point in $\pi(\ell, r')$, taking into account the current values of $q(i)$ over that path. Let $i^* = \mathrm{par}(i_{k^*})$. WDP queries the label of $i^*$: If $y(i^*) = 0$, WDP continues the binary search on $\pi(\ell, i_{k^*})$; if instead $y(i^*) = 1$, the binary search continues on $\pi(i^*, r')$, and so on. During the binary search, whenever WDP finds a node $u \in V(\pi(\ell, r'))$ with queried labels $y(u) = 0$ and $y(\mathrm{par}(u)) = 1$, then the edge of $\pi(\ell, r')$ cut by $c^*$ has been found, and the binary search on this backbone path terminates. In the special case where $y(r') = 1$, the binary search also ends, and we know that all nodes in $L(r')$ form a cluster of $\mathcal{C}(c^*)$. Once a binary search terminates, WDP updates $F$ and the probabilities $\boldsymbol{p}(i)$ at all nodes $i$ in the subtrees of $F$, so as to reflect the new knowledge gathered by the queried labels.

Below, we prove WDP's query complexity. The proofs are split into a series of lemmas.

**Lemma 2** *Given tree $T$ with set of leaves $L$, any prior $\mathbb{P}(\cdot) \in \mathcal{P}_{>0}$ over $c^*$, and any $c^* \sim \mathbb{P}(\cdot)$, let $j_0$ be a node of $\mathcal{AB}(c^*)$, having as children a leaf $\ell \in L$ and an internal node $j'$ of $T$ (see Figure 6, left). Then, during its execution, WDP will never select the bottom-up path starting from $\ell$.*

*Proof.* Let $T_0$ be the tree made up of all nodes of $\mathcal{AB}(c^*)$, and consider any given round with $q(i)$ in (3) defined by the current *posterior* distribution maintained by the algorithm. We first show that, for all ancestors $a$ of $j_0$, path $\pi(\ell, a)$ cannot be selected by WDP, because its entropy[5] $H(\pi(\ell, a))$ will always be strictly smaller than $H(\pi(\ell', a))$ for *all* leaves $\ell' \in L(j')$. To this effect, we can write

$$
H(\pi(\ell, a)) - H(\pi(\ell', a)) = \left( -q(\ell) \log_2 q(\ell) - \sum_{u \in \pi(j_0, a)} q(u) \log_2 q(u) \right)
$$
$$
- \left( H(\pi(\ell', j')) - \sum_{u \in \pi(j_0, a)} q(u) \log_2 q(u) \right)
$$
$$
= -q(\ell) \log_2 q(\ell) + \sum_{v \in \pi(\ell', j')} q(v) \log_2 q(v) . \tag{5}
$$

Now, since

$$
\sum_{u \in \pi(\ell', j')} q(u) + \sum_{v \in \pi(j_0, a)} q(v) = \sum_{u \in \pi(\ell', a)} q(u) = \sum_{u \in \pi(\ell, a)} q(u) = q(\ell) + \sum_{u \in \pi(j_0, a)} q(u) ,
$$

we have $q(\ell) = \sum_{v \in \pi(\ell', j')} q(v)$.

Consider the function $f(x) = -x \log_2 x$, for $x \in [0, 1]$. For all $x, x_1, x_2 \in (0, 1)$ such that $x_1 + x_2 = x$, it is easy to verify that we have $f(x) < f(x_1) + f(x_2)$. More generally, for all $x, x_1, x_2, \ldots, x_m \in (0, 1)$ with $\sum_{i=1}^{m} x_i = x$, one can show that $f(x) < \sum_{i=1}^{m} f(x_i)$. Since $|V(\pi(\ell', j'))| \geq 2$ (holding because $j' \notin L$ implies $\ell' \neq j'$), the above inequality on $f(\cdot)$ allows us to write

$$
-q(\ell) \log_2 q(\ell) < - \sum_{v \in \pi(\ell', j')} q(v) \log_2 q(v) ,
$$

i.e., (5) < 0. Notice that the assumption $\mathbb{P}(\cdot) \in \mathcal{P}_{>0}$ implies $q(v) > 0$ at any stage of the execution of WDP where node $v$ has an unrevealed label. This is because, after any binary search on a path selected by WDP, for all $v$ belonging to any tree in $F$, in the update phase each value $q(v)$ is multiplied by a strictly positive value. This ensures that we can use the above inequality about $f(\cdot)$, as its argument will always lie in the open interval $(0, 1)$.

The inequality in (5) implies that there always exists a leaf $\ell'$ of $T(j')$ such that WDP selects the path connecting $\ell'$ with the root of the tree containing $\ell$ in the current forest $F$. This selection entails the disclosure of either cut edge $(j', j_0)$ (if $j_0 \in L(T_0)$), or a cut edge in $T(j')$ (if $j_0 \notin L(T_0)$), which in turn implies that the labels of $i_0$ and all its ancestors will be disclosed to the algorithm to be equal to 0, thereby indirectly revealing also cut edge $(\ell, i_0)$. Since $F$ contains only trees whose height is larger than 1, after this cut edge disclosure the tree made up of leaf $\ell$ alone cannot be part of $F$, thus preventing WDP's selection of a path starting from $\ell$. $\square$

**Figure 6:** Illustration of all possible cases of Lemma 2 and Lemma 3. Nodes belonging to $T_0$ (see main text) are black, all remaining nodes are white. Leaves and subtrees of $T$ are represented by squares and triangles, respectively. Each node of $T'_{c^*}$ is enclosed in a circle. **Above:** The two possible cases illustrating Lemma 2, that is, $j' \notin V(T_0)$ on the left, and $j' \in V(T_0)$ on the right. **Below:** The five cases described in Lemma 3.

**Lemma 3** *For any input tree $T$ and any cut $c^*$ with $\mathbb{P}(\cdot) \in \mathcal{P}_{>0}$, the number of paths selected by* WDP *before stopping is $\widetilde{K}(T, c^*)$.*

*Proof.* If $c^*$ has only one cluster the statement is clearly true, since the binary search performed by WDP on the first selected path reveals that $y(r) = 1$ (hence $y(v) = 1$ for all $v \in V$). We then continue by assuming $y(r) = 0$, so that $c^*$ has least two clusters.

Let $\Pi$ be the set of all paths selected by WDP during the course of its execution. The binary search perfomed by WDP on *each* of such paths, discloses exactly *one* edge of $c^*$. Let $c^*_{wdp}$ be the set containing all these cut edges, and $c^*_0$ be the set of the remaining cut edges of $c^*$. We show that, for any $T$ and any cut $c^*$ of $T$, $|c^*_{wdp}| = \widetilde{K}(T, c^*)$, while all edges in $c^*_0$ are *indirectly* disclosed by WDP, although none of them belongs to paths in $\Pi$.

Let $T_0$ be the subtree of $T$ made up of all nodes in $\mathcal{AB}(c^*)$. The edges of $c^*$ can be partitioned into the five disjoint sets $S_1, \ldots, S_5$ (see Figure 6 for reference):

$S_1$: The set of all *pairs* of edges connecting a leaf $\ell_0$ of $T_0$ to two sibling leaves $\ell_1$ and $\ell_2$ of $T$ (Figure 6, below, 1);

$S_2$: The set of all *pairs* of edges connecting a leaf $\ell_0$ of $T_0$ to two sibling internal nodes $i_1$ and $i_2$ of $T$ (Figure 6, below, 2);

$S_3$: The set of all *pairs* of edges connecting a leaf $\ell_0$ of $T_0$ to a leaf $\ell \in L$ and an internal node $i$ of $T$ (Figure 6, below, 3);

$S_4$: The set of all edges connecting an internal node $i_0$ of $T_0$ to an internal node $i$ of $T$, so that the sibling node of $i$ belongs to $V(T_0)$ (Figure 6, below, 4);

$S_5$: The set of all edges connecting an internal node $i_0$ of $T_0$ to a leaf $\ell$ of $T$, so that the sibling node of $\ell$ belongs to $V(T_0)$ (Figure 6, below, 5).

Recall that $T'_{c^*}$ is the subtree of $T$ whose nodes are $(\mathcal{AB}(c^*) \cup \mathcal{LB}(c^*)) \setminus L$, and that $\widetilde{K}(T, c^*)$ is the number of its leaves. The leaves of $T'_{c^*}$ can be partitioned into the following four sets $A$, $B$, $C$, and $D$ (see again Figure 6 for reference):

$A$: The set of all leaves of $T'_{c^*}$ that are also leaves of $T_0$, i.e., that belong to $\mathcal{AB}(c^*)$;

$B$: The set of all *sibling* leaves of $T'_{c^*}$ that are also (sibling) internal nodes of $T$;

$C$: The set of all leaves of $T'_{c^*}$ that are also internal nodes of $T$ such that their sibling node is a leaf of $T$;

$D$: The set of all leaves of $T'_{c^*}$ that are also internal nodes of $T$ such that their sibling node belongs to $T_0$.

We will not show a one-to-one mapping between $L(T'_{c^*})$ and the cut edges of $c^*_{wdp}$ covering all possible cases.

$S_1 \leftrightarrow A$: For each pairs of cut edges in $S_1$), WDP clearly selects a path starting from either $\ell_1$ or $\ell_2$, which will indirectly disclose the cut edge incident to the sibling leaf ($\ell_2$ or $\ell_1$, respectively). $S_1$ is therefore about all leaves of set $A$.

$S_2 \leftrightarrow B$: For each pairs of cut edges in $S_2$, WDP selects two paths, one per cut edge. Each of these two paths clearly contains one of these two cut edges, and corresponds to all leaves of $T'_{c^*}$ that are also leaves of $T$. Hence we are covering all leaves of set $B$.

$S_3 \leftrightarrow C$: For the edges in $S_3$, WDP selects only one path, starting from a leaf of $T(i)$. This path clearly contains edge $(i, \ell_0)$, and covers all leaves of set $C$. Observe that, by Lemma 2, edge $(\ell_0, \ell)$ is always indirectly revealed and never contained in a path selected by WDP.

$S_4 \leftrightarrow D$: For the edges in $S_4$, whenever WDP selects a path starting from a leaf of $T(i)$, all the nodes in $V(T(i))$ are indirectly labeled 1, and from that point on, they will not be included in a tree in $F$. This path clearly contains edge $(i_0, i)$, hence we are covering all leaves of set $D$.

$S_5 \leftrightarrow \emptyset$: Finally, Lemma 2 ensures that all cut edges in $S_5$ are indirectly disclosed whenever WDP selects a path starting from a leaf belonging to $T(i'_0)$, where $i'_0$ is the sibling node of $\ell$. Hence this case is ruled out by Lemma 2, and does not correspond to any leaf.

From the above, we conclude that the number of paths selected by $wdp$ is always equal to $\widetilde{K}(T, c^*)$, as claimed. $\square$

The next lemma provides an entropic bound on the (condionally) expected number of queries WDP makes on a given path. Notice that the posterior distribution maintained by WDP never changes *during* each binary search, but only between a binary search and the next. Consider then $q(i)$ defined in (3) at the beginning of a given binary search in terms of the current posterior distribution, and let $\pi$ be the selected path, after having observed the labels that generated the current posterior.

**Lemma 4** *Let $\pi$ be any path selected by* WDP *during the course of its execution, and $\{q(i)\}$ be the current distribution (3) at the time $\pi$ is selected. Then the expected number of queries* WDP *makes on $\pi$, conditioned on past revealed labels, is $\mathcal{O}\left(\lceil H(\pi)\rceil\right) = \mathcal{O}\left(\log(|V(\pi)|)\right)$. Here, both the conditional expectation and $H(\pi)$ are defined i.t.o. $\{q(i)\}$.*

*Proof.* Let $\pi$ be the currently selected path, and denote by $E_{\text{par}}(\pi)$ the set made up of the edges in $\pi$ along with the edge connecting the top node of $\pi$ to its parent (recall that in the special case where $r$ is a terminal node of $\pi$, we can view $r$ as the child of a dummy "super-root"). The binary search performed on $\pi$ guarantees that the number of queries $Q(\pi, (u, v))$ made by WDP to find a cut edge $(u, v)$ lying on $\pi$ can be quantified as follows:

$$\left\lceil \log_2 \left( \frac{\sum_{(u', v') \in E_{\text{par}}(\pi)} \mathbb{P}((u', v') \in c^*)}{\mathbb{P}((u, v) \in c^*)} \right) \right\rceil = \left\lceil \log_2 \left( \frac{1}{\mathbb{P}((u, v) \in c^*)} \right) \right\rceil ,$$

where the probabilities above are defined w.r.t. the posterior distribution at the beginning of the binary search. The expected number of queries made on $\pi$, conditioned on past labels can thus be bounded as

$$\sum_{(u', v') \in E_{\text{par}}(\pi)} \mathbb{P}((u', v') \in c^*) \left\lceil \log_2 \left( \frac{1}{\mathbb{P}((u', v') \in c^*)} \right) \right\rceil = \sum_{u \in V(\pi)} q(i) \left\lceil \log_2 \left( \frac{1}{q(i)} \right) \right\rceil$$
$$= \mathcal{O}\left(\lceil H(\pi)\rceil\right)$$
$$= \mathcal{O}\left(\log(|V(\pi)|)\right) ,$$

as claimed. $\square$

We are now ready to prove Theorem 3 and Theorem 6.

*Proof.*[Theorem 3] For given $c^*$, let $\Pi = \Pi(c^*) = \langle \pi_1, \ldots, \pi_{|\Pi|} \rangle$ be the sequence of paths selected by WDP, sorted in the temporal order of selection during WDP's run. Also, denote by $Q(\pi_j)$ the number of queries made by WDP on $\pi_j \in \Pi$. Notice that the sequence $\Pi$ is fully determined by $c^*$. Moreover, the paths in $\Pi$ are orderer in such a way to guarantee that $\pi_j$ contains a unique edge

$(\mathrm{par}(u_j), u_j)$ which $c^*$ cuts across. Then, if we denote by $\{q_j(\cdot)\}$ the value of $q(\cdot)$ at the time path $\pi_j$ is selected, it is easy to see that cut $c^*$ can be alternatively generated by sequentially generating edge $(\mathrm{par}(u_1), u_1)$ according to distribution $\{q_1(\cdot)\}$ over $\pi_1$, then $(\mathrm{par}(u_2), u_2)$ according to (posterior) distrubution $\{q_2(\cdot)\}$ over $\pi_2$, then $(\mathrm{par}(u_3), u_3)$ according to (posterior) distrubution $\{q_3(\cdot)\}$ over $\pi_3$, and so on until $|\Pi|$ cuts have been generated. From Lemma 3, we have $|\Pi| = \widetilde{K}(T, c^*)$.

Let us then denote by $\mathbb{E}[\cdot]$ the expectation w.r.t. the *prior* distribution, and by $\mathbb{E}_j[\cdot]$ be the conditional expectation $\mathbb{E}[\cdot \mid (\mathrm{par}(u_1), u_1), (\mathrm{par}(u_2), u_2), \dots, (\mathrm{par}(u_{j-1}), u_{j-1})]$. Notice that the sequence of random variables $\mathrm{par}(u_1), u_1, (\mathrm{par}(u_2), u_2), \dots, (\mathrm{par}(u_{j-1}), u_{j-1})$ fully determines the posterior distribution $\{q_j(\cdot)\}$ before the selection of the $j$-th path $\pi_j$, and so, $\pi_j$ itself. This way of viewing $c^*$ makes $\widetilde{K} = \widetilde{K}(T, c^*)$ a (finite) stopping time w.r.t. the sequence of random variables $\mathrm{par}(u_1), u_1, (\mathrm{par}(u_2), u_2), \dots,$, in that $\{\widetilde{K} \geq j\}$ is determined by $(\mathrm{par}(u_1), u_1), (\mathrm{par}(u_2), u_2), \dots, (\mathrm{par}(u_{j-1}), u_{j-1})$. This allows us to write

$$
\begin{aligned}
\mathbb{E}\left[\sum_{j=1}^{\widetilde{K}} Q(\pi_j)\right] &= \sum_{i=1}^{n}\sum_{j=1}^{i} \mathbb{E}\left[Q(\pi_j)\{\widetilde{K}=i\}\right] \\
&= \sum_{j=1}^{n} \mathbb{E}\left[Q(\pi_j)\{\widetilde{K}\geq j\}\right] \\
&= \sum_{j=1}^{n} \mathbb{E}\left[\{\widetilde{K}\geq j\}\mathbb{E}_j[Q(\pi_j)]\right] \qquad \text{(since } \widetilde{K} \text{ is a stopping time)} \\
&= \sum_{j=1}^{n}\sum_{i=j}^{n} \mathbb{E}\left[\{\widetilde{K}=i\}\mathbb{E}_j[Q(\pi_j)]\right] \\
&= \sum_{i=1}^{n}\sum_{j=1}^{i} \mathbb{E}\left[\{\widetilde{K}=i\}\mathbb{E}_j[Q(\pi_j)]\right] \\
&= \sum_{i=1}^{n} \mathbb{E}\left[\{\widetilde{K}=i\}\sum_{j=1}^{\widetilde{K}}\mathbb{E}_j[Q(\pi_j)]\right] \\
&= \mathbb{E}\left[\sum_{j=1}^{\widetilde{K}}\mathbb{E}_j[Q(\pi_j)]\right] \\
&= \mathcal{O}\left(\mathbb{E}\left[\sum_{j=1}^{\widetilde{K}}\lceil H_j(\pi_j)\rceil\right]\right) \qquad \text{(by Lemma 4, where } H_j(\cdot) \text{ is w.r.t. } \{q_j(\cdot)\}) \\
&\qquad\qquad\qquad\qquad\qquad\qquad\qquad\qquad\qquad\qquad\qquad\qquad\qquad (6) \\
&= \mathcal{O}\left(\mathbb{E}[\widetilde{K}]\log h\right),
\end{aligned}
$$

as claimed $\square$

A slightly more involved guarantee for WDP is given by the following theorem, where the query complexity depends in a more detailed way on interplay between $T$ and the prior $\mathbb{P}(\cdot)$. Given any bottom-up path $\pi$ in $T$, we denote by $\widetilde{H}(\pi)$ the *normalized* entropy of $\pi$, defined as $\widetilde{H}(\pi) = -\sum_{i \in V(\pi)} \widehat{q}(i)\log(\widehat{q}(i))$, where $\widehat{q}(i) = q(i)/\sum_{i \in V(\pi)} q(i)$, and $q(i)$ is defined according to the prior distribution $\mathbb{P}(\cdot)$, as in (3). Notice that we may have $\sum_{i \in V(\pi)} q(i) < 1$. Further, denote by $\mathbb{D}$ the family of all sets $\Pi$ of all vertex-disjoint bottom-up paths starting from $T$'s leaves such that the top terminal node of each path $\pi \in \Pi$ is either the root $r$ of $T$ or a node of another path of $\Pi$. The upper bound in the following theorem is *never* worse than the upper bound in Theorem 3.

**Theorem 6** *In the noiseless realizable setting, for any tree $T$, any prior distribution $\mathbb{P}(\cdot)$ over $c^*$ such that $\mathbb{P}(\cdot) \in \mathcal{P}_{>0}$, the expected number of queries made by* WDP *to find $c^*$ is* $\mathcal{O}\left(\max_{\Pi \in \mathbb{D}} \sum_{j=1}^{m(\Pi)} \lceil \widetilde{H}(\pi_{i_j})\rceil\right)$ *, where $m(\Pi) = \min\left\{\left\lceil \mathbb{E}\widetilde{K}(T, c^*)\right\rceil, |\Pi|\right\}$, and paths $\pi_{i_1}, \pi_{i_2}, \dots$*

*in* $\Pi \in \mathbb{D}$ *are sorted in non-increasing value of normalized entropy* $\widetilde{H}(\cdot)$. *In the above, the expectations is again over* $\mathbb{P}(\cdot)$.

As an application of the above result, consider that oftentimes a linkage function generating $T$ also tags each internal node $i$ with a coherence level $\alpha_i$ of $T(i)$, which is typically increasing as we move downwards from root to leaves. A common situation in hierarchical clustering is then to figure out the "right" level of granularity of the flat clustering we are looking for through the definition of bands of nodes (i.e., bands of clusters) of similar coherence. This may be encoded through a prior $\mathbb{P}(\cdot)$ that uniformly spreads $(1 - \epsilon)/b$ probability mass over $b$-many edge-disjoint cuts of $T$, for $b \ll h$, and an arbitrarily small $\epsilon > 0$, and the remaining mass $\epsilon$ over all remaining cuts (this is needed to comply with the condition $\mathbb{P}(\cdot) \in \mathcal{P}_{>0}$). As we said in the main body of the paper, Theorem 6 gives a bound of the form $\mathbb{E}\widetilde{K}(T, c^*) \log b$ as opposed to the bound $\mathbb{E}\widetilde{K}(T, c^*) \log h$ provided by Theorem 3.

**Proof of Theorem 6**

*Proof.* Given $T$ and prior $\mathbb{P}(\cdot)$, let $\mathbb{D}_{wdp}$ be the set made up of all sets $\Pi$ of bottom-up paths in $T$ that WDP can potentially select during the course of its executions. Each set $\Pi$ is uniquely determined by $c^* \sim \mathbb{P}(\cdot)$. The family of sets $\mathbb{D}$ is clearly a superset of $\mathbb{D}_{wdp}$. We prove the theorem by showing that the expected number of queries made by WDP is upper bounded by

$$\mathcal{O}\left(\max_{\Pi \in \mathbb{D}_{wdp}} \sum_{j=1}^{m(\Pi)} \left\lceil \widetilde{H}(\pi_{i_j}) \right\rceil\right) , \tag{7}$$

where, for any given $\Pi \in \mathbb{D}_{wdp}$, $\pi_1, \pi_2, \ldots$ is the sequence of paths of $\Pi$ in the order they are selected by WDP, while $\pi_{i_1}, \pi_{i_2}, \ldots$ is the same sequence rearranged in non-increasing order of $\widetilde{H}(\cdot)$. Using the same notation as in the proof of Theorem 3, we observe that at the time when $\pi_j$ gets selected by WDP the distribution $\{q_j(\cdot)\}$ sitting along path $\pi_j$ is precisely the normalized distribution $\{\widehat{q}(\cdot)\}$ such that $\sum_{i=1}^{|V(\pi_j)|} \widehat{q}(i) = 1$, so that $H_j(\pi_j) = \widetilde{H}(\pi_j)$. Then, Eq. (6) combined with Lemma 4 allows us to write

$$\mathbb{E}\left[\sum_{j=1}^{\widetilde{K}} Q(\pi_j)\right] = \mathbb{E}\left[\mathcal{O}\left(\sum_{j=1}^{\widetilde{K}} \lceil \widetilde{H}_j(\pi_j) \rceil\right)\right] .$$

In the sequel, we show how to upper bound the right-hand side of the last (in)equality by (7). Set for brevity $\mathbb{E}[\widetilde{K}] = \lceil \mu \rceil$. We have

$$\sum_{j=1}^{\widetilde{K}} \lceil \widetilde{H}_j(\pi_j) \rceil = \sum_{j=1}^{\widetilde{K}} \{\widetilde{K} < \mu\} \lceil \widetilde{H}_j(\pi_j) \rceil + \sum_{j=1}^{\widetilde{K}} \{\widetilde{K} \geq \mu\} \lceil \widetilde{H}_j(\pi_j) \rceil$$

$$\leq \sum_{j=1}^{\mu} \lceil \widetilde{H}_j(\pi_{i_j}) \rceil + \frac{\widetilde{K}}{\mu} \sum_{j=1}^{\widetilde{K}} \lceil \widetilde{H}_j(\pi_j) \rceil$$

$$\leq \max_{\Pi \in \mathbb{D}_{wdp}} \sum_{j=1}^{m(\Pi)} \lceil \widetilde{H}_j(\pi_{i_j}) \rceil + \frac{\widetilde{K}}{\mu} \max_{\Pi \in \mathbb{D}_{wdp}} \sum_{j=1}^{m(\Pi)} \lceil \widetilde{H}_j(\pi_{i_j}) \rceil$$

$$= \left(1 + \frac{\widetilde{K}}{\mu}\right) \max_{\Pi \in \mathbb{D}_{wdp}} \sum_{j=1}^{m(\Pi)} \lceil \widetilde{H}_j(\pi_{i_j}) \rceil ,$$

so that, taking the expectation of both sides,

$$\mathbb{E}\left[\sum_{j=1}^{\widetilde{K}} \lceil \widetilde{H}_j(\pi_j) \rceil\right] \leq 2 \max_{\Pi \in \mathbb{D}_{wdp}} \sum_{j=1}^{m(\Pi)} \lceil \widetilde{H}_j(\pi_{i_j}) \rceil .$$

This concludes the proof. $\square$

## B.4 N-WDP (Noisy Weighted Dichotomic Path)

N-WDP is a robust variant of WDP that copes with persistent noise. Given an internal node $i \in V \setminus L$, let $\mathcal{L}(i)$ be the set of all possible queries that can be made to determine $y(i)$, i.e., the set $(\ell, \ell') \in L(\text{left}(i)) \times L(\text{right}(i))$. Then, given confidence $\delta \in (0, 1]$, and noise level $\lambda \in [0, 1/2)$, N-WDP:

1. Preprocesses $T$ and prior $\mathbb{P}(\cdot)$ by setting $y(i) = 1$ for all nodes $i \in V \setminus L$ such that $|\mathcal{L}(i)| < \frac{\alpha \log(n/\delta)}{(1-2\lambda)^2}$, for a suitable constant $\alpha > 0$. $\mathbb{P}(\cdot)$ is also updated (all $j \in T(i)$ have $\boldsymbol{p}(j) = 1$). At the end of this phase, each node in $V$ is either unlabeled or labeled with 1.

2. Let $T_\lambda$ be the subtree of $T$ made up of all unlabeled nodes of $T$, together with all nodes whose label has been set to 1 that are children of unlabeled nodes. N-WDP operates on $T_\lambda$ as WDP, with the following difference: Whenever a label $y(i)$ is requested, N-WDP determines its value by a majority vote over $\Theta\left(\frac{\log(n/\delta)}{(1-2\lambda)^2}\right)$-many queries selected uniformly at random from $\mathcal{L}(i)$.

**Proof sketch of Theorem 4**

*Proof.* Let $\Lambda$ be the set of pairs of leaves whose label has been corrupted by noise. A standard Chernoff bound implies that for any fixed subset of $L \times L$ containing at least $\alpha \frac{\log(1/\delta)}{(1-2\lambda)^2}$ pairs (for a suitable constant $\alpha > 0$), the probability that the majority of them belongs to $\Lambda$ is at most $\delta$. Let us set for brevity $f(n, \lambda, \delta) = \alpha \frac{\log(n/\delta)}{(1-2\lambda)^2}$. A union bound over the at most $n-1$ internal nodes of $V$ guarantees that for all queries $y(i)$ made by N-WDP operating on $T_\lambda$ the majority vote over $f(n, \lambda, \delta)$-many queries on pairs of leaves of $\mathcal{L}(i)$ will produce the correct label (i.e., before noise) of that node with probability at least $1 - \delta$.

Moreover, since the cut $\widehat{c}$ found by N-WDP on $T_\lambda$ can be obtained with probability at least $1 - \delta$ from $c^*$ by merging zero or more clusters on $T$, it is immediate to see that $\widetilde{K}(T_\lambda, \widehat{c}) \leq \widetilde{K}(T, c^*)$. It is also easy to verify that this inequality holds even in expectation over the prior distributions of cut $c^*$ on $T$ and $\widehat{c}$ on $T_\lambda$, that is, $\mathbb{E}_{\mathbb{P}_\lambda} \widetilde{K}(T_\lambda, \widehat{c}) \leq \mathbb{E}_{\mathbb{P}} \widetilde{K}(T, c^*)$, where $\mathbb{P}_\lambda$ denotes the modified prior on tree $T_\lambda$ produced after N-WDP's initial preprocessing (Step 1 in the main body of the paper).

Recall that, with probability $\geq 1 - \delta$, the behavior of $n - wdp$ on $T$ with prior $\mathbb{P}(\cdot)$ is the same as that of $wdp$ on $T_\lambda$ with the updated prior $\mathbb{P}_\lambda(\cdot)$. Then we can use Lemma 3 by replacing $c^*$ with $\widehat{c}$ to claim that the number of paths selected by N-WDP before stopping is $\widetilde{K}(T_\lambda, \widehat{c})$, and then Lemma 4 to conclude that the expected (*w.r.t.* $\mathbb{P}(\cdot)$) number of queries made by N-WDP is upper bounded with probability $1 - \delta$ (over the noise in the labels) by

$$\mathcal{O}\left(f(n, \lambda, \delta) \, \mathbb{E}_{\mathbb{P}_\lambda} \widetilde{K}(T_\lambda, \widehat{c}) \log(h(T_\lambda))\right) = \mathcal{O}\left(\frac{\log(n/\delta)}{(1-2\lambda)^2} \, \mathbb{E}_{\mathbb{P}} \widetilde{K}(T, c^*) \log h\right) .$$

We conclude the proof by showing that with probability at least $1 - \delta$ we have $d_H(\Sigma^*, \widehat{\mathcal{C}}) = \mathcal{O}\left(\frac{n(\log(n/\delta))^{3/2}}{(1-2\lambda)^3}\right)$. Since all labels requested by N-WDP are simultaneously correct with probability at least $1 - \delta$, the distance $d_H(\Sigma^*, \widehat{\mathcal{C}})$ is upper bounded with the same probability by $\sum_{i \in L(T_\lambda)} |L(i)|^2$. For each tree $T_\lambda$ constructed by N-WDP, and any $i \in L(T_\lambda)$, we have

$$\mathcal{O}(f(n, \lambda, \delta)) = |L(i)| = \Omega\left(f(n, \lambda, \delta)^{1/2}\right) .$$

Hence, the maximum number of leaves of $T_\lambda$ is $\mathcal{O}\left(\frac{n}{(f(n, \lambda, \delta))^{1/2}}\right)$, and the quantity $\sum_{i \in L(T_\lambda)} |L(i)|^2$, contributing to $d_H(\Sigma^*, \widehat{\mathcal{C}})$ is upper bounded by

$$\mathcal{O}(n \left(f(n, \lambda, \delta)\right)^{3/2}) = \mathcal{O}\left(\frac{n(\log(n/\delta))^{3/2}}{(1-2\lambda)^3}\right) ,$$

as claimed. $\square$

## C   Missing material from Section 4

### C.1   Proof sketch of Theorem 5

*Proof.* The proof follows from Theorem 2 and 3 in [6], together with the following observations.

1. For any tree $T$ with $n$ leaves, we have $|\mathbb{C}(T,K)| = \mathcal{O}(n^K)$.

2. When $\mathcal{D}$ is uniform, the disagreement coefficient $\theta = \theta(\mathbb{C}(T,K), \mathcal{D})$ is $\mathcal{O}(K)$. To show this statement, consider the following. For any $c^* \in \mathbb{C}(T,K)$ and $r > 0$, let

$$DIS(c^*, r) = \Big\{ (x_1, x_2) \in L \times L : \exists c' \in \mathbb{C}(T,K) : \sigma_{\mathcal{C}(c')}(x_1, x_2) \neq \sigma_{\mathcal{C}(c^*)}(x_1, x_2)$$
$$\wedge\, d_H(\Sigma_{\mathcal{C}(c')}, \Sigma_{\mathcal{C}(c^*)}) \leq r \Big\}.$$

Then in our case $\theta$ is defined as

$$\theta = \sup_{r > 0} \frac{|\{(x_1, x_2) \in DIS(c^*, r)\}|}{r\, n^2}.$$

Now, for any budget $r$ in $DIS(c^*, r)$, and any $c^* \in \mathbb{C}(T,K)$, the number of times we can replicate the perturbation of $c^*$ so as to obtain $c'$ satisfying $d_H(\Sigma_{\mathcal{C}(c')}, \Sigma_{\mathcal{C}(c^*)}) \leq r$ is at most $K$. This is because any such perturbation will involve a different cluster of $\mathcal{C}(c^*)$, and therefore disjoint sets of leaves. Moreover, each such perturbation covers $rn^2$ leaves. The worst case that makes $\theta = K$ is when $T$ is a full binary tree, and $\mathcal{C}(c^*)$ has equally-sized clusters. In all other cases $\theta \leq K$.

3. Regarding the expected running time per round, we give the pseudocode (see Algorithm 2 in this appendix) of a sequential algorithm, which operates as follows. In a preliminary phase the input tree $T$ is preprocessed in order to be able to find in *constant time* at any time $t$ **(i)** the leftmost and rightmost descendent leaf of any internal node of $T$, and **(ii)** the lowest common ancestor of any two given leaves.[6] At each time $t$, it receives $\langle (x_{i_t}, x_{j_t}), \sigma_t, w_t \rangle$, for some weight $w_t \geq 0$, and label $\sigma_t \in \{-1, +1\}$, and outputs $\mathrm{err}_t(\mathcal{C}(\hat{c}_{t+1}))$, based on the past computation of $\mathcal{C}(\hat{c}_t)$ and $\mathrm{err}_{t-1}(\mathcal{C}(\hat{c}_t))$. This can be directly used to compute at each round $\mathrm{err}_{t-1}(\mathcal{C}(\hat{c}_t))$ needed by the algorithm, but also the perturbed cut $\hat{c}'_t$ and its associated empirical error $\mathrm{err}_{t-1}(\mathcal{C}(\hat{c}'_t))$, once we repeat the computation by perturbing the last item $\langle (x_{i_t}, x_{j_t}), \sigma_t, w_t \rangle$ in the training set as follows: $\sigma_t = -\sigma_{\mathcal{C}(\hat{c}_t)}(x_{i_t}, x_{j_t})$, and $w_t = \infty$. In turn, the above can be used to compute $d_t = \mathrm{err}_{t-1}(\mathcal{C}(\hat{c}'_t)) - \mathrm{err}_{t-1}(\mathcal{C}(\hat{c}_t))$ and probability $p_t$.

The cornerstone of this procedure is to maintain updated over time for each internal node $v$ of $T$ a record storing eight values:

- *1st, 2nd, 3rd and 4th values*: positive and negative inter-cluster total weight of all leaves in $L(\mathrm{left}(v))$ and $L(\mathrm{right}(v))$;
- *5th and 6th values*: positive and negative inter-cluster sum of weights $w(x_i, x_j)$ for all $x_i \in L(\mathrm{left}(v))$ and all $x_j \in L(\mathrm{left}(v))$, and
- *7th and 8th values*: total intra-cluster negative weight of all the clusters of leaves in $L(\mathrm{left}(v))$ and $L(\mathrm{right}(v))$.

When this procedure receives in input triplet $\langle (x_{i_t}, x_{j_t}), \sigma_t, w_t \rangle$, it finds $a_t = \mathrm{lca}(x_{i_t}, x_{j_t})$. Then the eight records associated with each node on the bottom-up path $\pi(a_t, r)$ are updated in a bottom-up fashion according to the input, whenever necessary. This requires a constant time per node in $V(\pi(a_t, r))$. Finally, $\mathrm{err}_t(\mathcal{C}(\hat{c}_{t+1}))$ is obtained by simply summing the total intra-cluster negative weight of all clusters of leaves in $L(\mathrm{left}(r))$ and $L(\mathrm{right}(r))$ to the total inter-cluster positive weight of all leaves in $L(\mathrm{left}(r))$ and $L(\mathrm{right}(v))$, plus the inter-cluster sum of positive weights of the pairs $w(x_i, x_j)$ for all $x_i \in L(\mathrm{left}(r))$ and $x_j \in L(\mathrm{left}(r))$. In the special case where the updated clustering is made up of a single cluster containing all leaves of $T$, the procedure outputs the sum of all negative values in the record associated with $r$. In any event, computing this sum requires constant time.

Hence the total time required for performing all operations required at any time $t$ is simply $\mathcal{O}(|V(\pi(a_t, r))|)$.

This concludes the proof. $\square$

Finally, in order to compute the clustering at the end of the training phase, it suffices to perform a breadth-first visit of $T$ to find all leaves of $T'_{c^*}$. This requires a time linear in the number of clusters of the clustering found by the algorithm. Then the algorithm outputs the indices of the leftmost and rightmost descendant of each leaf of $T'_{c^*}$, which requires $\Theta(1)$ time per cluster. The total time for giving in output the computed clustering is therefore linear in the number of its own clusters.

## C.2 Pseudocode of the NR algorithm in the non-realizable setting

Each internal node of $T$ is associated with a record containing eight values that are maintained updated over time. We start by providing the semantics of these eight values:

- weight$(v, \text{left}, -1)$ and weight$(v, \text{left}, +1)$: negative and positive inter-cluster total weight of leaves in $L(\text{left}(v))$.

- weight$(v, \text{middle}, -1)$ and weight$(v, \text{middle}, +1)$: negative and positive inter-cluster sum of weights $w(\ell_l, \ell_r)$, where $\ell_l \in L(\text{left}(v))$ and $\ell_r \in L(\text{right}(v))$, respectively.

- weight$(v, \text{right}, -1)$ and weight$(v, \text{right}, +1)$: negative and positive inter-cluster total weight of leaves in $L(\text{right}(v))$.

- cost$(v, \text{left})$ and cost$(v, \text{right})$: intra-cluster total negative weight of clusters of leaves in $L(\text{left}(v))$ and $L(\text{right}(v))$, respectively.

Finally, for any internal node $v$ of $T$, we denote by $s(v)$ the following sum:

$$
\begin{aligned}
s(v) \overset{\text{def}}{=} &\ \text{weight}(v, \text{left}, -1) + \text{weight}(v, \text{left}, +1) + \text{weight}(v, \text{middle}, -1) \\
&+ \text{weight}(v, \text{middle}, +1) + \text{weight}(v, \text{right}, -1) + \text{weight}(v, \text{right}, +1) \ .
\end{aligned}
$$

**Algorithm 2:** Sequential algorithm for the non-realizable case (NR).

▷ **INPUT** : Sequence of pairs of labeled leaves of the form $\langle (\ell, \ell'), \sigma(\ell, \ell') \rangle$
▷ **OUTPUT**: Clustering $\mathcal{C}$ with minimum cost over all clusterings realized by $T$.

**Init:**

- **for** $v \in V$ **do**      **if** $v \in L$ is_cluster$(v) \leftarrow 1$; **else** is_cluster$(v) \leftarrow 0$;
- current_tot_cost $\leftarrow 0$;
- Preprocess $T$ in a bottom-up fashion and store for each internal node of $T$ the leftmost and rightmost leaf descendant index. /* Necessary to output $\mathcal{C}$ in linear time */
- Preprocess $T$ to find the lowest common ancestor of any pair of leaves in constant time.

**for** $t = 1$ **to** ... **do**
 Receive pair of leaves $(\ell, \ell')$;
 $w(\ell, \ell') \leftarrow 0$; /* initialize $w(\ell, \ell')$ */
 $a \leftarrow$ lowest common ancestor of $\ell$ and $\ell'$; /* we assume $\ell \neq \ell'$ */
 /* save all records for the rollback that will be done later */
 $\mathcal{S} \leftarrow$ list of saved records (eight values per node) of the path $\pi(a, r)$;

 /* ----- verify whether $\ell$ and $\ell'$ are in the same cluster of the current optimal clustering ----- */
 **while** $a \neq r \wedge$ is_cluster$(a) = 0$ **do**
  $\lfloor\ a \leftarrow$ par$(a)$;
 **if** is_cluster$(a) = 1$ **then** same_cluster$(\ell, \ell') \leftarrow 1$; **else** same_cluster$(\ell, \ell') \leftarrow 0$;

 /* ----- compute optimal cost under constraint ----- */
 **if** same_cluster$(\ell, \ell') = 1$ **then**
  /* compute the optimal cost of the current clustering constrained by the assumption that $\ell$ and $\ell'$ are in different clusters; $-\infty$ is simulated using a very large negative number */
  total_modified_cost $\leftarrow$add_weight$(\ell, \ell', -\infty)$;
 **else**
  /* compute the optimal cost of the current clustering constrained by the assumption that $\ell$ and $\ell'$ are in the same cluster; $+\infty$ is simulated using a very large positive number */
  total_modified_cost $\leftarrow$add_weight$(\ell, \ell', +\infty)$;
 /* rollback of the clustering preceding the add of weight $-/+\infty$ */
 Restore all records of $\mathcal{S}$;

 /* ----- add weight $w(\ell, \ell')$ if necessary ----- */
 Set:

 - Difference $d_t \leftarrow \frac{1}{t-1}$ (total_modified_cost $-$ current_tot_cost) ;
 - Probability $p_t$ as a function of $d_t$ as in Eq. (4);
 - $w(\ell, \ell') \leftarrow \frac{\sigma(\ell, \ell')}{p_t}$;
 - With probability $p_t$, current_tot_cost $\leftarrow$add_weight$(\ell, \ell', w(\ell, \ell'))$;

/* ----- find the current optimal clustering/partition of $L$ ----- */
Perform a breadth-first search on $T$, starting from its root $r$, to create the set $V'$ formed by all nodes $v \in V$ such that is_cluster$(v) = 1$ and for all ancestors $a$ of $v$ we have is_cluster$(a) = 0$;

$\mathcal{C} \leftarrow \emptyset$;
**for** $v \in V'$ **do**
 $\lfloor\ \mathcal{C} \leftarrow \mathcal{C} \cup \{L(v)\}$;
**return** $\mathcal{C}$ .

---

**Procedure** Procedure `add_weight`$(\ell, \ell', w(\ell, \ell'))$

---

▷ **INPUT**   : Pair of leaves $\ell, \ell' \in L$ (with $\ell \neq \ell'$) and weight $w(\ell, \ell')$
▷ **OUTPUT**: Total clustering cost after adding weight $w(\ell, \ell')$

$a \leftarrow$ lowest common ancestor of $\ell$ and $\ell'$;

```
/*  update middle weight record of node a  */
```
$\text{weight}(a, \text{middle}, \text{sgn}(w(\ell, \ell'))) \leftarrow \text{weight}(a, \text{middle}, \text{sgn}(w(\ell, \ell'))) + w(\ell, \ell')$;

```
/*  set cluster flag of node a  */
```
**if** $s(a) \geq 0$ **then**
   |   is_cluster$(a) \leftarrow 1$;
**else**
   |__ is_cluster$(a) \leftarrow 0$;

```
/* ----- record update of all a's ancestors ----- */
```
**while** $a \neq r$ **do**

```
/*  set variable dir to left or right direction from par(a) to a  */
```
   **if** $a = \text{left}(\text{par}(a))$ **then**
      |   dir $\leftarrow$ left;
   **else**
      |__ dir $\leftarrow$ right;

```
/*  update positive and negative inter-cluster weights of node par(a)  */
```
   **for** $\sigma \in \{+1, -1\}$ **do**
      **if** is_cluster$(a) = 0$ **then**
         |   $\text{weight}(\text{par}(a), \text{dir}, \sigma) \leftarrow$
         |   $\text{weight}(a, \text{left}, \sigma) + \text{weight}(a, \text{middle}, \sigma) + \text{weight}(a, \text{right}, \sigma)$;
      **else**
         |__ $\text{weight}(\text{par}(a), \text{dir}, \sigma) \leftarrow 0$;

```
/*  update par(a)'s cost record relative to node a  */
```
   **if** is_cluster$(a) = 0$ **then**
      |   $\text{cost}(\text{par}(a), \text{dir}) \leftarrow \text{cost}(a, \text{left}) + \text{cost}(a, \text{right})$;
   **else**
      |   $\text{cost}(\text{par}(a), \text{dir}) \leftarrow \text{cost}(a, \text{left}) + |\text{weight}(a, \text{left}, -1)| + |\text{weight}(a, \text{middle}, -1)| +$
      |__ $|\text{weight}(a, \text{right}, -1)| + \text{cost}(a, \text{right})$;

```
/*  update cluster flag of par(a)  */
```
   **if** $s(\text{par}(a)) \geq 0$ **then**
      |   is_cluster$(\text{par}(a)) \leftarrow 1$;
   **else**
      |__ is_cluster$(\text{par}(a)) \leftarrow 0$;
   |__ $a \leftarrow \text{par}(a)$;

```
/* ----- compute the total cost of the current optimal clustering ----- */
```
**if** is_cluster$(r) = 0$ **then**
   |   cost_after_adding_weight $\leftarrow \text{cost}(r, \text{left}) + \text{weight}(r, \text{left}, +1) + \text{weight}(r, \text{middle}, +1) +$
   |   $\text{weight}(r, \text{right}, +1) + \text{cost}(r, \text{right})$
**else**
   |   cost_after_adding_weight $\leftarrow \text{cost}(r, \text{left}) + |\text{weight}(r, \text{left}, -1)| +$
   |__ $|\text{weight}(r, \text{middle}, -1)| + |\text{weight}(r, \text{right}, -1)| + \text{cost}(r, \text{right})$;

**return** cost_after_adding_weight .

---

## C.3   Missing material from Section 5

In Table 2 we report the results of our preliminary experiments. Notice that N-WDP, NR, and BF are randomized algorithms. Hence, for these three algorithms we give average results and

| Tree | No. of queries<br>Algorithm | 250 | 500 | 1000 | 2000 | 5000 | 10000 | 20000 |
|---|---|---|---|---|---|---|---|---|
| SING | ERM | 8.81 | 8,78 | 8.39 | 8.29 | 8.29 | 8.29 | 8.29 |
| | N-WDP | 8.29±0.0 | 8.28±0.0 | 8.28±0.0 | 8.29±0.0 | – | – | – |
| | NR | 11.0±2.0 | 8.77±0.0 | 8.43±0.0 | 8.31±0.0 | 8.29±0.0 | – | – |
| | BF | 89.0±0.0 | 89.0±0.0 | 88.0±0.0 | 86.0±2.0 | 87.0±3.0 | 72.0±10.0 | 67.0±10.0 |
| MED | ERM | 10.30 | 10.16 | 9.36 | 8.91 | 8.91 | 8.69 | 8.65 |
| | N-WDP | 9.41±0.1 | 9.07±0.1 | 8.88±0.1 | 8.92±0.1 | 8.8±0.1 | 8.8±0.1 | 8.7±0.1 |
| | NR | 10.17±0.0 | 9.37±0.0 | 9.0±0.0 | 8.85±3.0 | – | – | – |
| | BF | 89.4±0.0 | 88.1±0.0 | 87.0±0.0 | 63.1±0.0 | 18.2±5.0 | 18.0±3.0 | 10.9±1.0 |
| COMP | ERM | 10.65 | 10.30 | 10.04 | 9.26 | 9.06 | 8.99 | 8.93 |
| | N-WDP | 9.52±0.0 | 9.47±0.0 | 9.44±0.0 | 9.43±0.0 | – | – | – |
| | NR | 10.1±0.0 | 10.0±0.0 | 10.0±0.0 | 11.4±0.6 | 10.8±0.5 | 9.0±0.0 | 8.9±0.0 |
| | BF | 13.5±0.0 | 13.5±0.0 | 9.2±0.0 | 9.1±0.0 | 9.0±0.0 | 9.0±0.0 | 8.9±0.0 |

**Table 2:** Test error (in percentage) vs. number of queries for the various algorithms we tested on the hierarchies SING, MED, and COMP originating from the MNIST dataset (see main body of the paper). Standard deviations are also reported. Missing values on N-WDP are due to the fact that the algorithm stops before reaching the desired number of labels. Missing values on NR are instead due to the fact that we stopped the algorithm's execution once we obseved no further test error improvement.

standard deviation across 10 independent runs of each one of them. As a reference, consider that the performance of BEST (see Section 5 in the main body of the paper) on the three datasets is the following: SING: 8.26%, MED: 8,51%, COMP: 8.81%. Moreover, since in this dataset we have 10 class labels with approximately the same frequency, both a *random* clustering and a degenerate clustering having $n = 10000$ singletons would roughly give 10% error.

In light of the above, notice that on both SING and MED, the robust breadth-first strategy BF goes completely off trail, in that it tends to produce clusterings with very few clusters. This behavior is due to the presence in the two hierarchies of long paths starting from the root, which is in turn caused by the way the single and the median linkage functions deal with the outliers contained in the MNIST dataset.

Finally, one should take into account the fact that when training our active learning algorithms we have used the first 500 labels for parameter tuning. Hence, a fair comparison to ERM is one that contrasts the test error of N-WDP, NR, and BF at a given number of queries $q$ to the test error of ERM at $q + 500$ queries. From Table 2 one can see that, even with this more careful comparison, N-WDP outperforms ERM. On the other hand, NR looks similar to ERM on MED and COMP, and worse than ERM on SING.