[Reviews · NeurIPS 2019]

Reviewer 1



The supplement seems to be an exact copy of the manuscript. The manuscript is very difficult to read and understand. To improve readability, the manuscript should be organized into Algorithm, Theorem and examples. The main flaw of this paper is the lack of experiment and insightful interpretation. One of the main appealing points in this paper is the speed, thus it should be addressed in the experiment. This lack makes the paper’s quality and significant decreasing. line 46: why do you need to split the version space evenly?

Reviewer 2



This paper considers a setting in which we are given some hierarchical clustering over data, and we want to find some pruning/cut of this hierarchy by adaptively querying for must-link/cannot-link constraints among its leaves. There are three settings considered in this paper, each with its own algorithms and analysis: (i) the noiseless setting, where each queried pair is consistent with some true cut of the tree, (ii) the persistent noise setting, where each queried pair is consistent with some true cut of the tree except for some randomly selected subset of pairs for which the response is flipped, and (iii) the agnostic setting, in which no assumptions are made over the distribution of responses. The paper presents active learning algorithms for each of these settings, demonstrating that the number of queries can be upper bounded by quantities that depend on the tree and possibly the ground truth clustering. Some of these bounds are also accompanied by nearly-tight lower bounds on the query complexity of any active learning algorithm. In all, this paper represents a very thorough investigation into this problem, and I vote for acceptance. The settings considered here are quite diverse, and the algorithms are both practical and have rigorous query complexity and running time guarantees. The comparison to related work is also quite thorough, showing improvement over the bounds given in more generic active learning algorithms. Minor comments: - The notation is a little difficult to follow, possibly due to the overloading of notation. In particular, defining the prior distribution over cuts as P(c) in terms of the values P(i) makes certain statements confusing. Typos: - Line 22: obiquitous

Reviewer 3



This paper derives complexity results for active learning queries to hierarchical clustering. The result is a partition or "cut", c, of the cluster tree, where the "flat" clustering is defined by the clusters at the leaves of a subtree of nodes AB(c) that have the same root as the original cluster tree. Learning occurs by making pairwise judgments on items (leaf nodes). All pairwise judgments form a "ground truth" matrix \Sigma. Given consistency conditions, this is an equivalent way to represent a clustering. The objective of the learning process (e.g. "accuracy") not highlighted in the paper; the so called "good cut" mentioned p.3 l. 124, however the measure of success appears to be the ability to learn a clustering consistent with the \Sigma matrix. This raises a concern: A consistent or realizable \Sigma implies one possible clustering out of all those generated by cuts on one cluster tree. The Error Measure proposed (p. 4 l. 139) simply computes a distance from the one \Sigma - defined clustering. So \Sigma has no sense of a clustering "close" in the tree to the one it represents. How then can \Sigma be used to determine the "right level of granularity of the flat clustering? (p.7, l 273). In the so called "non realizable" case, apparently assuming an arbitrary \Sigma matrix, the paper builds on methods of Importance weighted active learning (IWAL). Ironically, it appears that any practical active learning setting will fall into the non-realizable case, and here the results are limited so far. . How one might exploit the methods proposed in the paper deserves further thought. Active learning presupposes an individual, an actual person comes to mind, whose effort one intends to conserve. The generation of pairwise queries on top of a set of priors assigned to nodes in T as a form of cluster learning, given one already has an agglumerative clustering may not be an efficent way to do active learning. Given the approach taken, the development of ideas in the paper is tedious, and the results appear to be modest, with shallow and not deep insight Typos: p.1 l. 36 We are lead -> we are led p. 2, l 55. by adaptating -> by adapting

Reviewer 4



Originality: I believe the authors have introduced novel approaches for selecting a flat clustering from a hierarchical clustering. I believe that this work is interesting and novel. Some closely related work is only mentioned in the supplementary material and I think it would be beneficial to move to the body of the paper with further discussion. Particularly, the line of work on clustering with same/different cluster membership queries [Wang et al 2012], [Vesdapunt et al, 2014], [Firmani et al, 2016] (others mentioned in this paper). Particularly, I'd be very interested to better understand the tradeoffs between the assumptions of this work and the related work and to better understand the motivations for the assumptions of this paper. I believe that the proposed work requires fewer queries, but at the same time is limited in selecting a tree consistent partition. Other related work that I think should be more thoroughly discussed includes: active learning methods for building the hierarchy by querying for pairwise similarities [Eriksson et al, 2011] and [Krishnamurthy et al, 2012] and the constraint-based approach of Vikram & Dasgupta [2016]. In these cases feedback is used to construct the hierarchy. While some use pairwise similarities, I imagine the works could be extended to think about those similarities as 1/0 labels for same cluster / different cluster. It would be insightful to understand / highlight advantages / disadvantages of the two approaches-- particularly regarding the fact that the desired clustering I believe is always realizable in the related work. Quality: The approaches in the paper seem to be technically sound. My main concerns about quality regard the problem setting and how being restricted to a tree-consistent partition is limited. I think a conclusion should be added to the paper. Clarity: The paper is clearly written. Clarifications on the above related work would be beneficial. While I realize that space is limited, I think more detail on the non-realizable case would be helpful-- perhaps some kind of illustration that shows for a ground truth clustering C* several tree structures for which C* is not realizable and for each cut that minimizes the error measure. Significance: I believe the authors contribute results that are important to the area of work on active/interactive clustering. If the authors were to provide additional motivation for problem setting, I think that this would be very beneficial. In particular, when would it be preferable to do gather feedback for selecting a cut from an existing tree rather than gathering feedback in building the tree itself (and say using that feedback to select a cut). [Wang et al 2012] Jiannan Wang, Tim Kraska, Michael J. Franklin, and Jianhua Feng. "Crowder: Crowdsourcing entity resolution." Proceedings of the VLDB Endowment 5, no. 11 (2012): 1483-1494. [Vesdapunt et al, 2014] Norases Vesdapunt, Kedar Bellare, and Nilesh Dalvi. "Crowdsourcing algorithms for entity resolution." VLDB 2014 http://www.vldb.org/pvldb/vol7/p1071-vesdapunt.pdf [Firmani et al, 2016] Donatella Firmani, Barna Saha, and Divesh Srivastava. "Online entity resolution using an oracle." Proceedings of the VLDB Endowment 9, no. 5 (2016): 384-395. http://www.vldb.org/pvldb/vol9/p384-firmani.pdf [Eriksson et al, 2011] Brian Eriksson, Gautam Dasarathy, Aarti Singh, and Rob Nowak. Active Clustering: Robust and Efficient Hierarchical Clustering using Adaptively Selected Similarities. AISTATS. 2011. http://proceedings.mlr.press/v15/eriksson11a/eriksson11a.pdf [Krishnamurthy et al, 2012] Akshay Krishnamurthy, Sivaraman Balakrishnan, Min Xu, and Aarti Singh. "Efficient active algorithms for hierarchical clustering." ICML (2012). https://icml.cc/2012/papers/473.pdf [Vikram & Dasgupta, 2016] Sharad Vikram, and Sanjoy Dasgupta. "Interactive bayesian hierarchical clustering." ICML. 2016.

[Author Response · NeurIPS 2019]

1　We thank all Reviewers for their helpful comments; specific responses are given below.

2　**Reviewer 1**

3　1. *Supplement exact copy of manuscript :* No. Beyond the main body, the supplement contains 15 pages of appendices,
4　with pseudocode, proofs of theorems, and the results of our preliminary experiments (Sect. C.3) in a tabular form.

5　2. *Main flaw is lack of experiment ... speed should be addressed:* Our experiments are a preliminary validation of our
6　theoretical results (the latter being the main meat of the paper). Despite we used only one dataset, we intentionally
7　generated 3 very diverse input trees out of it (LL. 325-344, and Tables 1 and 2), so as to enlarge the scope of this
8　empirical validation. As for speed, please observe that our theoretical results come with running time analyses.

9　3. *line 46: even split of version space:* Even split of version space (whenever possible) is a standard baseline approach
10　to active learning [12,24,14,15,26,23].

11　**Reviewer 2**

12　1. *Notation to improve:* Thanks for pointing this out, we'll find a better notation so as to avoid notation overloading.

13　**Reviewer 3**

14　1. *On motivation of our problem:* It is often the case in big organizations that data processing pipelines are split
15　into *services*, making Machine Learning solution providers be constrained by the existing hardware/software
16　infrastructure. In the Active Learning (AL) applications that motivate this work, the hierarchy over the items to be
17　clustered *is not build by us*, it is rather provided by a third party, i.e., an exogenous data processing tool that relies
18　on side information on the items (e.g., word2vec mappings and associated distance functions) which are possibly
19　generated by yet another service, etc. In this modular environment, it is reasonable to assume that the tree $T$ is *given*
20　*to us as part of the input* of our AL problem. The human feedback our algorithms rely upon may or may not be
21　consistent with the tree $T$ at hand both because human feedback is generally noisy and because this feeback may
22　originate from yet another source of data, e.g., another production team in the organization that was not in charge of
23　building the original tree. In fact, the same tree over the data items may serve the clustering needs of different groups
24　within the organization, having different goals and views on the same data. This also motivates why we are led to
25　consider different noise scenarios (realizable, noisy realizable, and nonrealizable). In short, we are not artificially
26　constraining ourselves to clusterings realized by a tree, since the tree is itself part of the input. Our preliminary
27　experiments have perhaps been a bit misleading, in that the trees we managed to generate ourselves.
28　The above can be added to the paper to better motivate our investigation.

29　2. *On the objective of the learning process and clustering closeness:* We are not $100\%$ sure we fully understood this
30　comment. If $\Sigma$ (the human feedback) is consistent with $T$ (realizable setting) then being close in Hamming distance
31　to the underlying clustering represented by $\Sigma$ implies being close to that clustering in any "reasonable" sense. In
32　particular, zero Hamming distance implies the two clusterings coincide. On the other hand, if $\Sigma$ is not realized by $T$,
33　then we are either in the noisy realizable or in the nonrealizable setting, where the goal is different (like bounding the
34　excess risk in Eq. (2)). On the contrary, if the reviewer is alluding to using a distance metric other than Hamming,
35　e.g., one that depends on the structure of $T$, this is a relatively easy adaptation of what is currently in the paper. The
36　very the reason why we restricted ourselves to Hamming distance (over matrices) stems from our need to treat in a
37　unified manner both the (noisy) realizable and the nonrelizable settings, since some such alternative distances would
38　only apply to the realizable setting (with or without noise), but not to the nonrealizable one with i.i.d. entries.

39　3. *It appears that any practical ... fall into the non-realizable case:* This statement looks too broad to be considered
40　undisputed ... As a striking counterexample, one of the findings of our experiments (see Appendix C.3) is that
41　AL algorithms assuming persistent noise can in practice be more effective than those making the more general
42　nonrealizable assumption. Notice that in our experiments $\Sigma$ has been generated by the MNIST class labels, hence $\Sigma$
43　has virtually nothing to do with the trees we generated, which in turn do not rely on those labels at all. This offers a
44　hopefully clearer interpretation of our empirical findings, which we can further elaborate upon in the paper.

45　**Reviewer 5**

46　1. *On motivation:* Please see response 1 to Reviewer 3.

47　2. *Related work and tradeoffs:* Thanks for bringing the first three references to our attention, we shall duly compare to
48　them. In the time frame of this rebuttal, what we can say is that it seems like these papers are not readily comparable
49　to ours, since the noise assumptions are slightly different, e.g., the uneven noise gap analysis in Firmani et al. (Thm
50　1 and 2), and Prop. 3-5 therein. Anyhow, the reviewer is right in that, being restricted to a clustering realized by
51　$T$, we generally need less queries. For instance, combining our Thms. 2 and 3, our bound in the realizable case
52　is between $\mathbb{E}[K]$ and $\mathbb{E}[K] \log h$, while in general it takes $n\mathbb{E}[K]$ queries to fully reconstruct the clustering [13].
53　Also recall the lower bound $n - K + \binom{K}{2}$ in Wang et al., and Firmani et al. As for the other papers the reviewer is
54　mentioning, we'll definitely cite them, but they seem to be facing the problem of building the hierarchy, rather than
55　cutting a pre-existing one (by the way, another reference along this line is [11]). It is currently unclear to us which
56　comparison can be made.

57　3. *More details in the non-realizable case:* Thanks for your suggestions, we'll add more illustrations.

[Meta-Review · NeurIPS 2019]

This paper studies an interesting theoretical question of clustering using two data sources: a hierarchy and interactive queries on link constraints. The reviewers appreciate the fact that the algorithm can achieve sharp query complexity guarantees under challenging noisy settings. The only weakness of the paper is motivation - what is a practical scenario where we have these two sources of data?